# HPV Vaccination in the U.S. Midwest: Barriers and Facilitators of Initiation and Completion in Adolescents and Young Adults

**DOI:** 10.3390/vaccines13111175

**Published:** 2025-11-20

**Authors:** Kristyne D. Mansilla Dubon, Edward S. Peters, Shinobu Watanabe-Galloway, Abraham Degarege

**Affiliations:** Department of Epidemiology, College of Public Health, University of Nebraska Medical Center, Omaha, NE 68198, USA; kmansilladubon@unmc.edu (K.D.M.D.); epeters@unmc.edu (E.S.P.); swatanabe@unmc.edu (S.W.-G.)

**Keywords:** HPV vaccine uptake, Midwest, HPV vaccination predictors

## Abstract

**Background/Objectives**: HPV vaccination uptake among adolescents and young adults in the US remains low, and coverage in the Midwest falls short of the Healthy People 2030 goal of 80%. **Methods**: A cross-sectional survey of adolescents and young adults was conducted to identify facilitators and barriers to HPV vaccination uptake among adolescents and young adults in the Midwest. **Results**: Out of 1306 individuals aged 13–26 years, 397 (30.4%) were fully vaccinated (2–3 doses), 124 (9.5%) had received one dose, 324 (24.8%) were unvaccinated, and 461 (35.3%) were unsure of their vaccination status. Awareness of HPV vaccines (OR: 2.4, 95% CI: 1.6, 3.6), beliefs about vaccine effectiveness (OR: 1.8, 95% CI: 1.1, 2.9), family support (OR: 2.3 95% CI: 1.4, 3.8) and knowing someone with cervical cancer (OR: 1.8, 95% CI: 1.2, 2.7) were associated with increased odds of full vaccination. Beliefs in vaccine safety (OR: 2.0, 95%CI: 1.0, 3.9) and having health insurance coverage (OR: 1.9, 95% CI: 1.0, 3.5) were associated with increased odds of initiated vaccination (i.e., receiving at least one dose). Concerns about vaccine side effects (OR: 0.5, 95% CI: 0.3, 0.8) and not receiving recommendations from doctors were significantly associated with decreased odds of full vaccination (OR: 0.5, 95% CI: 0.3, 0.8) or initiated vaccination (OR: 0.5% CI: 0.2, 0.9). Clinician recommendations and awareness also reduced the likelihood of unknown vaccination status. Race-stratified analyses suggested heterogeneity in predictors across racial/ethnic groups. **Conclusions**: Our findings support the need for multi-level interventions aimed at increasing HPV vaccination initiation and completion in the Midwest.

## 1. Introduction

Human papillomavirus (HPV) is the most common sexually transmitted infection in the United States, with an estimated prevalence of ~40% among young women and men aged 15 to 24 years [1]. In 2018, around 42 million Americans were infected with HPV types that are associated with cancer and other diseases [2,3,4,5,6]. Most HPV infections clear spontaneously [7], but non-oncogenic types can cause benign lesions such as cutaneous genital warts, while infection with oncogenic HPV types is the main cause of cervical cancer and anal cancer [8]. HPV has also been associated with vaginal (~75%), vulvar (~69%), penile (~63%), and oropharyngeal cancers (~70%) [5,7,8,9]. HPV-related cancers can be prevented through vaccination [2,10,11,12]. The U.S. Centers for Disease Control and Prevention’s (CDC) Advisory Committee on Immunization Practices (ACIP), along with the American Academy of Pediatrics (AAP) and the American Cancer Society (ACS), recommend initiating HPV vaccination at ages 11–12 years with the option to begin as early as age 9 and providing catch-up vaccination through age 26 [13,14]. Individuals who start vaccination before age 15 should receive 2 doses, spaced 6–12 months apart, and those who start vaccination after age 15 or are immunocompromised should receive a three-dose series [14].

Despite efforts to expand vaccine coverage, HPV vaccination uptake among adolescents and young adults in the US remains low [15]. Coverage among adolescents aged 13–17 years was 62.9% in 2024 [15,16], and among young adults, 47% in 2022 [17,18,19]. Sex differences have been observed in both groups: in young adults, coverage was 58% for females and 40% for males; in adolescents, 65% for females and 61% for males. More specifically, in the Midwest region, coverage among adolescents aged 13–17 years ranged from 58% to 78% in 2022 [20]. Although coverage in some states in the Midwest remains higher than the national average, it still falls short of the Healthy People 2030 goal of 80% [21]. Studies have identified factors such as geographic, racial, gender, and age disparities as reasons for the decreased uptake of the HPV vaccine in the US [22,23,24,25,26,27,28]. Individual factors such as attitudes, perceived social expectations to get vaccinated, beliefs of self-ability to accomplish vaccination, and perceived risks of HPV-related disease have been shown to influence vaccination intention [29,30,31,32,33,34,35,36,37]. Other reported barriers include lack of knowledge about HPV and the HPV vaccine, concerns regarding its safety and efficacy, financial constraints, and fears of discrimination associated with receiving the vaccine [34,38,39,40,41]. Additionally, newer studies have found areal, societal, and policy-level factors influencing vaccination uptake [22,34]. In the post-COVID-19 era, growing public skepticism toward immunization programs has renewed attention to HPV vaccine hesitancy, highlighting the need to understand changing attitudes and confidence in vaccination [17,24,42].

Most studies that have examined behavioral, cultural, and other predictors of HPV vaccination focused on age- or gender-specific populations [19,31,33,35,41,43]. Additionally, many of these studies have relied on small sample sizes, specific age and gender groups, or were conducted outside of the US [39,44,45,46,47]. To the best of our knowledge, few studies have explored HPV vaccination predictors at the state level within the Midwest region of the US [48,49]. Even fewer studies have been conducted at the regional level across the Midwest, where vaccination remains suboptimal according to the National Immunization Survey, but regional data availability is limited [20,50,51]. The Midwest region of the US is home to diverse racial and ethnic groups, and it is characterized by vast rural areas where limited healthcare access and higher poverty rates contribute to poorer health outcomes [24,52]. In the U.S. Midwest, HPV vaccination is administered primarily through pediatricians, family physicians, local health departments, community health centers, and Vaccines for Children (VFC) providers, following CDC and Advisory Committee on Immunization Practices (ACIP) guidelines. Unlike countries with national school-based programs, HPV vaccination in the United States, including all Midwestern states, is delivered through opportunistic, clinic-based systems. While some Midwestern states (e.g., Michigan, Minnesota, and Illinois) have school-linked or outreach vaccination events, none operate a universal school-based program. This decentralized approach contributes to variability in initiation and completion across counties and sociodemographic groups, as access depends on healthcare utilization and provider recommendation rather than systematic in-school delivery. The Midwest HPV vaccine initiation and completion rates have been consistently reported as the second lowest among US regions [53,54]. Further research is needed in the region to elucidate barriers to HPV vaccination among adolescents and young adults, identify individual and group-level patterns of vaccine uptake, and explore perceptions of HPV, HPV-related cancers, and factors influencing vaccination decisions. This study examined a comprehensive range of factors influencing HPV vaccine uptake, guided by the Integrated Health Theory [24,52], Theory of Planned Behavior [53], Health Behavior Theory [54,55,56,57], and prior related research [58,59,60,61,62]. These theoretical frameworks suggest that individuals’ intentions to get vaccinated are shaped by their attitudes, social influences, and confidence in their ability to act, while their beliefs and background affect these attitudes indirectly. In this context, vaccine hesitancy may be the result of how people think and feel about vaccines, shaped by their knowledge, experiences, and trust in the health system. Moreover, psychosocial drivers (i.e., confidence in vaccine safety, perceived disease risk, social norms, and trust in providers) influence initiation of HPV vaccination, while cultural beliefs and historical mistrust may affect confidence across communities. Structural barriers such as access, cost, and continuity of care also influence HPV vaccine completion, emphasizing that both psychological trust and systemic convenience are necessary for sustained vaccination coverage [55,56]. Furthermore, the digital environment may play a role in vaccine uptake by increasing misinformation about HPV, HPV-related cancer, and the vaccine among the target population [57]. The aim of this study was to identify facilitators and barriers to HPV vaccination uptake among adolescents and young adults in the Midwest region of the US and to examine whether these facilitators and barriers differed across racial subgroups within this population. We hypothesized that attitudes, beliefs, and social influences would be key predictors of vaccination intentions and behaviors and that these determinants might vary by race and cultural context. By identifying key barriers and facilitators, the study seeks to inform evidence-based, culturally responsive strategies to improve HPV vaccination coverage and equity in public health practice.

## 2. Methods

### 2.1. Study Population

We conducted a cross-sectional survey of adolescents and young adults aged 13 to 26 years old, living in the Midwest region of the United States (i.e., Illinois, Ohio, Michigan, Indiana, Missouri, Wisconsin, Minnesota, Kansas, Iowa, Nebraska, South Dakota, and North Dakota) between 28 March and 17 April in 2023. The Midwestern region had a combined population of approximately 69 million residents in 2023. According to American Community Survey (ACS) estimates, adolescents aged 13–17 years represent roughly 7 to 8% of the total population in the region, and young adults aged 18 to 26 years represent 11–12%. Across states in the region, the population distributions among adolescents and young adults were close to parity [58].

### 2.2. Procedures

We administered a questionnaire from March 2023 to April 2023 through Qualtrics online research panel services, a commercial survey sampling and administration company in the US (Qualtrics, Provo, UT, USA) [63]. Briefly, Qualtrics recruited participants to respond to this questionnaire using a combination of actively managed, double-opt-in market research panels. The sample was chosen from a preestablished pool of participants who had agreed to be contacted for research purposes. Panelists first completed a standardized questionnaire in Qualtrics to generate demographic and behavioral profiles, which were then used to randomly select eligible respondents for survey participation. Eligible participants received an invitation via email to participate in the online survey, which did not contain information about the details of the survey, to avoid self-selection bias. After reading the informed consent and assenting to participate, Qualtrics screened respondents for age and state of residence. Afterwards, if eligible, the respondents were directed to the online questionnaire. Sampling was stratified by sex (approximately 50% female and 50% male) and age group (13–17 years: ~45%; 18–22 years: ~40%; 23–26 years: ~15%) across the 12 Midwestern United States. Moreover, participants were compensated for their participation.

Although using an online panel may introduce self-selection bias, Qualtrics mitigates this risk through actively managed, double-opt-in recruitment, demographic quota sampling, and rigorous quality controls (e.g., attention checks, IP validation, and digital fingerprinting). Weighting procedures were applied when necessary to align the sample with population benchmarks, thereby enhancing representativeness and data reliability [59,60,61].

### 2.3. Measures

The questionnaire collected data on knowledge, attitude, and practices about HPV vaccination and included 58 questions guided by the Integrated Health Theory [24,52], the Theory of Planned Behavior [53], the Health Behavior Theory [54,55,56,57], and prior related research [58,59,60,61,62]. The survey also included questions about the respondent’s HPV vaccination uptake and sociodemographic characteristics, which included age, sex, race, ethnicity, education, state of residence, and information about other vaccines participants had received in their lifetime.

### 2.4. Outcome

Our outcome of interest was HPV vaccination uptake. For our analysis, we categorized HPV vaccination into four categories: those who reported having 2 or 3 doses as “fully vaccinated”, those who received one dose as “initiated vaccination”, and those who reported not having received any doses as “unvaccinated”. If respondents did not recall their vaccination status, their vaccination status was determined as “unknown.” We included all three vaccination uptake categories (initiated, fully vaccinated, and unvaccinated) and included those with unknown vaccination status as a separate category in the analysis. Our operational definition of “fully vaccinated” reflects the U.S. HPV vaccination schedule in place during the study period.

### 2.5. Exposure

We assessed 44 questions as independent predictors, which included awareness and knowledge about HPV infection, HPV-related cancers, and the HPV vaccine; beliefs about vaccination in general; beliefs about HPV vaccination; beliefs about susceptibility and severity of HPV infection; normative beliefs; self-efficacy; and cues to action for HPV vaccination on a five-point Likert scale as ‘strongly disagree’, ‘disagree’, ‘neutral’, ‘agree’, and ‘strongly agree’. Responses of ‘strongly disagree’ collapsed with ‘disagree’ and ‘strongly agree’ with ‘agree’ during data analysis. In this context, neutral meant neither agreeing nor disagreeing with the questions or statements presented in the survey. Responses to questions about cues to action, knowledge, and awareness of HPV were measured and analyzed as binary variables. We assessed awareness with 2 questions that inquire about having heard of HPV infection, HPV-related cancers, and the HPV vaccine.

We categorized self-reported race as non-Hispanic (NH) Black, NH White, NH Asian, American Indian/Alaska Native, Native Hawaiian/Other Pacific Islander, and Multiracial, with ethnicity (Hispanic/Latino) captured separately to define non-Hispanic race groups. We collapsed racial groups with a small sample size to preserve power (i.e., Other).

### 2.6. Sample Size

Since this study was part of a larger project evaluating the utility of health behavior theories in explaining factors influencing HPV vaccination using structural equation modeling (SEM), the sample size was determined based on the requirements of the overall study [50]. Briefly, the SEM based on the Integrated Health Theory (IHT) included 184 estimated parameters, comprising 39 factor loadings, 65 variances, 25 covariances, and 55 structural paths. A participant-to-parameter ratio of 7 is generally recommended to ensure adequate statistical power for SEM estimation [62]. Accordingly, a minimum of 1288 participants was required to test the validity of the conceptual frameworks derived from the IHT, Theory of Planned Behavior, and Health Belief Theory. Qualtrics ultimately provided data from 1306 participants, meeting this requirement.

### 2.7. Ethical Considerations

The Institutional Review Board (IRB) of the University of Nebraska Medical Center (IRB # 0696-22-EP) authorized this study. Because this survey posed a minimal risk to the respondents who were reached from existing pools of research panel lists of data processor companies, the IRB waived parental or guardian consent/permission for teenagers. Participants provided assent by checking a box after reading a consent form outlining the study’s purpose, confidentiality, risks, and benefits. Only individuals who gave assent proceeded to complete the survey. Qualtrics provides proprietary incentives based on survey length, participant profile, and completion rate. Participants who answer at least 50% of questions receive compensation, which may include cash, points, gift cards, or other rewards.

### 2.8. Statistical Analysis

We summarized the proportion and frequency of HPV vaccination uptake, sociodemographic variables, and the HPV independent predictors. Next, we examined bivariate associations between our outcome and sociodemographic variables using a chi-square test. Given the non-normal distribution of age, we assessed age differences across vaccination groups using the Kruskal–Wallis test. We identified predictors of HPV vaccination uptake using multinomial logistic regression analysis. We used stepwise model selection (at *p* = 0.05), including all 44 predictors plus sociodemographic variables (age, race, sex, and educational attainment) to adjust for potential confounding. We set unvaccinated and neutral answers to each predictor as reference values. Lastly, to examine heterogeneity in our results, we stratified the stepwise-selected model by racial groups.

We calculated prevalence odds ratios and 95% confidence intervals; furthermore, we calculated AIC, −2Log L, and likelihood ratio for both models to analyze model fit statistics. In order to reduce the chance of finding falsely significant results due to multiple comparisons, corrections were made by deploying the Benjamini–Hochberg method [63,64,65]. We performed analyses using SAS^®^ 9.4 (SAS Institute Inc., 2017, Cary, NC, USA) and created visualizations in R version 4.3.1 using (“ggplot2”) [66].

## 3. Results

We surveyed 1306 adolescents and young adults. We observed that 397 (30.4%) reported being fully vaccinated (having received two or three doses of the HPV vaccine), 124 (9.5%) had received only one dose, 324 (24.8%) were not vaccinated, and 461 (35.3%) did not know their vaccination status. The mean age of the 1306 participants was 19 years. Just over half (56%) of the study participants were female; this proportion remained consistent across all four groups (i.e., fully vaccinated, initiated vaccination, unvaccinated, and unknown vaccination status). Most participants identified as NH White (52%), NH Black or African American comprised 20% of our sample, and 17% reported Hispanic ethnicity. Close to 40% of participants reported having less than a high school education, while 27% had some college or higher educational attainment (Table 1).

In Table 2, we summarized data on the 44 independent predictors and the distribution of survey responses for each predictor, stratified by vaccination status. Overall, 70% of participants had heard of HPV infection, about 65% had heard of the HPV vaccine, and 75% had heard of HPV-related cancers. Notably, 84% understood that HPV infection can cause HPV-related cancers, and around 72% knew that these cancers are preventable through vaccination. However, 30% were concerned about vaccine side effects, and only 13% believed they were at risk of contracting HPV. More than half (approximately 50–60%) considered HPV infection and HPV-related cancers to be severe. Additionally, 37% reported that a doctor or healthcare provider had recommended they receive the HPV vaccine, while half of all participants believed that the vaccine was both safe and effective. Lastly, 55–58% reported they would be motivated to get vaccinated if they knew a woman with cervical cancer or a family member who had developed an HPV-related cancer.

Table 3 presents the results of the multinomial logistic regression model produced by stepwise model selection. The final model that predicted vaccination retained twelve of the forty-four independent predictors. After applying the FDR adjustment (q < 0.05), all predictors that were significant in the main multinomial logistic model remained statistically significant after multiple-testing correction.

### 3.1. Full Vaccination Predictors

Awareness of the HPV vaccine was associated with increased odds of full vaccination (OR: 2.4, 95% CI: 1.6, 3.6). In contrast, concerns about vaccine side effects were associated with decreased odds of full vaccination (OR: 0.5, 95% CI: 0.3, 0.8). Likewise, not receiving doctor or healthcare provider recommendations to get the HPV vaccine decreased the odds of full vaccination by 50% (OR: 0.5, 95% CI: 0.3, 0.8). Yet, receiving recommendations or support from a family member to receive the HPV vaccine was associated with higher odds of full vaccination status. (OR: 2.3 95% CI: 1.4, 3.8). Believing that the HPV vaccine is effective (OR: 1.8, 95% CI: 1.1, 2.9) and knowing someone with cervical cancer (OR: 1.8, 95% CI: 1.2, 2.7) were associated with an increased odds of full vaccination status.

### 3.2. Initiated Vaccination Predictors

Awareness of the HPV vaccine (OR: 2.4, 95% CI: 1.5, 4.0), beliefs that the HPV vaccine is safe (OR: 2.0, 95% CI:1.0, 3.9), and knowing someone with cervical cancer (OR: 2.4, 95% CI: 1.4, 3.9) were associated with increased odds of initiated vaccination status. Additionally, lack of health insurance coverage to receive the HPV vaccine was associated with increased odds of initiated vaccination status (OR: 1.9, 95% CI: 1.0, 3.5). On the other hand, not receiving doctor or healthcare provider recommendations was associated with lower odds of initiated vaccination (OR: 0.5% CI: 0.2, 0.9).

### 3.3. Unknown Vaccination Predictors

Appendix A shows the results for those who responded that they did not know their vaccination status. Consistent with results for fully and initiated vaccination participants, awareness about the HPV vaccine was associated with decreased odds of unknown vaccination status (OR: 0.6, 95% CI: 0.4, 0.8). Receiving doctor or healthcare provider recommendations to get the HPV vaccine reduced the odds of unknown vaccination by 40% (OR: 0.6% CI: 0.4, 0.9), compared to those who were unvaccinated. These findings suggest that social influence may have impact on vaccination decisions (Appendix A).

### 3.4. Race and Ethnic-Specific Regression Models

#### 3.4.1. Non-Hispanic Black and African American

Appendix A summarize findings on the stratified stepwise-selected model. Among non-Hispanic Black and African American participants in our study, we found that believing that the HPV vaccine is safe increased the odds of full vaccination (OR: 8.6, 95% CI: 2.4–30.3). However, concerns about vaccine side effects (OR: 0.4, 95% CI: 0.1, 0.9), believing one is too young to get vaccinated (OR: 0.2, 95% CI: 0.1, 0.9) decreased the odds of full vaccination status (OR: 0.3, 95% CI: 0.1, 0.9). HPV vaccine awareness (OR: 3.4, 95% CI: 1.1, 11.0), knowing someone with cervical cancer increased the odds of initiating vaccination (OR: 6.5, 95% CI: 2.0, 21.1). Some predictors that increased the odds of initiated vaccination status serve as barriers, meaning they increase the odds of initiated but incomplete vaccination (e.g., lack of health insurance coverage (OR: 4.3, 95% CI: 1.0, 18.2), disbelief about HPV cancer’s seriousness, (OR: 17.1, 95% CI: 2.3, 125.0) and disbelief that the HPV vaccine’s safety (OR: 6.4, 95% CI: 1.3, 32.3). Additionally, disbelief in the HPV vaccine’s effectiveness (OR: 0.2, 95% CI: 0.03, 0.8) and disagreement with the statement: “Vaccine is one way to ensure good health” decreased the odds of full vaccination status (OR: 0.3, 95% CI: 0.1–0.9) and decreased the odds of initiated vaccination. HPV-related cancers awareness decreased the odds of unknown vaccination (OR: 0.3, 95% CI: 0.2, 0.7).

#### 3.4.2. Non-Hispanic White

HPV vaccine awareness (OR: 3.0, 95% CI: 1.6, 5.7), believing that the HPV vaccine is effective (OR: 2.9, 95% CI: 1.4, 6.2), and family support or recommendations to receive the HPV vaccine (OR: 2.4, 95% CI: 1.1, 5.2) increased the odds of full vaccination. In contrast, concerns about vaccine side effects (OR: 0.5, 95% CI: 0.2, 0.9) and not receiving family support or recommendations to receive the HPV vaccine (OR: 0.3, 95% CI: 0.1, 0.7) decreased the odds of full vaccination. Both believing (OR: 3.3, 95% CI: 1.1, 9.3) and disbelieving (OR: 6.8, 95% CI: 2.1, 23.0) that HPV-related cancers are serious and disbelieving that the HPV vaccine is safe (OR: 3.4, 95% CI: 1.1, 10.6) increased the odds of initiating but not completing vaccination.

#### 3.4.3. Hispanic and Other Races

Knowing someone with cervical cancer increased the odds of full vaccination (OR: 3.2, 95% CI: 1.2, 8.3) and initiated vaccination (OR: 62.4, 95% CI: 2.8–>999). Not receiving family support or recommendations to receive the HPV vaccine (OR: 0.23, 95% CI: 0.1, 0.9) decreased the odds of full vaccination among Hispanic or Latino participants.

HPV-related cancer awareness (OR: 7.5, 95% CI: 1.2, 47.2) was associated with increased odds of full vaccination and with decreased odds of unknown vaccination status (OR: 0.2, 95% CI: 0.03, 0.9) among participants who reported other races.

## 4. Discussion

Our study identified individual, interpersonal, and organizational barriers and facilitators of HPV vaccination among both adolescents and young adults in the Midwest. These factors have been previously reported in other contexts but have not yet been comprehensively described in this region. Additionally, we observed race-specific patterns in barriers and facilitators. Only one-third of our study participants had a full vaccination status, contrary to recent reports about the national average estimate for adolescents (~78%) and young adults (~47%) [15,18,19,51], and far short of the Healthy People 2030 goal of 80% [17].

Facilitators associated with higher odds of full vaccination included HPV vaccine awareness, belief in vaccine effectiveness, provider recommendation, family support, and knowing someone with cervical cancer. Belief in vaccine safety was associated with vaccine initiation (≥1 dose). Provider recommendations and awareness of HPV-related cancers also reduced the likelihood of unknown vaccination status.

On the other hand, concerns about vaccine side effects were a significant barrier to full vaccination. Also, not receiving recommendations from doctors receiving the HPV vaccine, disbelief about the seriousness of HPV-related cancers and HPV vaccine safety were barriers to initiating vaccination. Lack of health insurance was associated with higher odds of vaccine initiation, suggesting a potential barrier to vaccine completion.

These findings are consistent with other studies in the US that have identified associations of attitudes, subjective norms, perceived susceptibility, perceived benefits, and beliefs about vaccine safety with an increased HPV vaccination uptake status [25,28,30,37,38,39,47,66]. Similarly to our findings, limited knowledge of HPV-related risks and low awareness have been repeatedly identified as barriers to HPV vaccination in other populations [39,44,45]. Moreover, factors such as health insurance coverage and trust in the effectiveness and safety of the HPV vaccine have been described as key facilitators of HPV vaccination among women, adolescents, and young adults globally [30,66].

Our results are in line with previously reported facilitators and barriers to HPV vaccination. We identified facilitators that may influence vaccination hesitancy among adolescents and young adults at the interpersonal and organizational levels. Our findings are consistent with other studies where provider recommendations have been significant facilitators to HPV vaccination initiation and completion [31,32,39,67]. Additionally, family support to receive the HPV vaccine and increased parental engagement in preventive health have also been described as being associated with increased vaccination completion [32,68]. Knowing someone with cervical cancer or HPV has been documented as a motivator of HPV vaccination among adult women [69]. Lastly, we found that concerns about vaccine side effects were also a barrier to both vaccine completion and initiation; however, studies that have identified barriers to HPV vaccination rarely describe concerns about vaccine side effects as a barrier at the individual level. Nonetheless, there are studies that describe this as a parental concern [40,45,70]. Finally, our results about disbelief about HPV-related cancers’ seriousness are consistent with literature reports of low perceived HPV-related cancer risks and low perceived HPV infection seriousness associated with no vaccination and lower vaccination intention [59,71,72,73].

In our race-stratified models, we observed that NH Black participants had more barriers (*n* = 8) to vaccination initiation and completion compared to NH White participants (*n* = 4); both racial groups had a similar number of facilitators (NH Black = 5 and NH White = 4), suggesting some heterogeneity in our results. However, we found that HPV vaccine awareness and beliefs about vaccine effectiveness were consistent facilitators of vaccine uptake among NH White and NH Black participants, but their roles were different between races. For NH White participants, HPV vaccine awareness and belief about vaccine effectiveness served as facilitators for full vaccination, and for NH Black participants, they facilitated vaccine initiation. Concerns about vaccine side effects, disbelief about the HPV vaccine’s safety, and the seriousness of HPV-related cancers were consistent barriers for vaccine initiation among both NH White and NH Black participants. The stratified analysis also showed that knowing someone with cervical cancer was a consistent facilitator for initiation and full vaccination among Hispanic/Latino participants and for vaccine initiation among NH Black participants. Our stratified models showed that not receiving family support to receive the HPV vaccine was a common barrier to full vaccination among NH White participants and Hispanic/Latino participants. Interestingly, we found that disbelief in vaccine effectiveness, thinking about being too young to receive the vaccine, and the belief that vaccines are not a way to ensure good health were barriers for vaccine uptake only among NH Black participants. Both Hispanic/Latino and Other had small sample sizes, yielding wide confidence intervals, and this prevented us from finding meaningful comparable differences among these racial and ethnic groups. However, these findings are hypothesis-generating and provide an avenue for further research.

Lastly, our findings align with and extend established behavioral models of vaccine hesitancy [55,56]. Consistent with the World Health Organization’s 5C framework, concerns about vaccine safety and side effects may reflect low confidence, whereas awareness, belief in vaccine effectiveness, and family support may reflect high confidence and collective responsibility. Perceived seriousness of HPV-related cancers addresses complacency, and healthcare coverage reflects the constraints dimension. These findings contribute to theoretical progress by showing how these factors work together to influence whether adolescents and young adults start and complete the HPV vaccination.

Our study enrolled a large, demographically balanced sample size that we consider broadly representative of the U.S. Midwest. Unlike many prior analyses that assessed adolescents and young adults separately, we examined both groups together to provide a more comprehensive view of vaccination coverage and predictors that may influence vaccination uptake. Additionally, we examined differences across racial and ethnic groups. The limitations of this study include potential selection bias inherent in the cross-sectional study design. In addition, we did not receive information about the number of eligible and contacted individuals in this study from the third party who administered the survey, which limited the evaluation of the generalizability of the study findings to the Midwest population. Furthermore, self-reported vaccination information is not validated by medical records, which can introduce information bias and measurement bias in our outcome. The cross-sectional nature of the study also limited the ability to establish causal relationships between knowledge, attitude, and practice predictors and vaccination uptake status. Additionally, our study was underpowered for group effects; future research should include oversampling of underrepresented racial groups to obtain precise estimates. Our study findings are restricted to the Midwest and may not be generalizable to the other regions in the US. Although our study identified key psychological and social facilitators and barriers to HPV vaccination, we did not assess broader contextual factors such as socio-digital inequalities that may limit access to reliable health information or the role of misinformation, which is increasingly recognized as a major contributor to vaccine hesitancy, particularly among adolescents in the digital era [45,55,70,74,75,76,77]. Moreover, our study refers to the Healthy People 2030 target of 80% HPV vaccination coverage as the current U.S. benchmark; however, it falls short of the WHO cervical cancer elimination framework’s 90% goal. This highlights the importance of continued regional progress to first meet national standards and ultimately advance toward global elimination targets.

Our study results revealed key areas for improvement to boost HPV vaccination rates among adolescents and young adults. More specifically, communication strategies focused on increasing awareness of HPV risks and vaccine safety may address key barriers identified in our study. Because facilitators of HPV vaccination extend beyond individual-level factors, multilevel interventions that target adolescents and young adults, families, and healthcare providers are warranted. Parent- and provider-focused initiatives that strengthen recommendation quality and educational messaging could increase uptake [55,75]. Expanding affordable access to HPV vaccines (e.g., through health insurance) can mitigate cost-related vaccine hesitancy. The descriptive patterns shown in Table 2, our findings highlight opportunities to improve public health messaging, particularly the need to strengthen physician recommendations, address concerns about vaccine safety and side effects, and emphasize both personal susceptibility to HPV and the preventive benefits of vaccination, which remain underrecognized among a sizable proportion of respondents.

A majority of respondents were aware of HPV and its link to cancer, yet a substantial proportion expressed concerns about vaccine safety and perceived low personal risk. These gaps emphasize the critical role of healthcare providers in recommending the HPV vaccine and addressing persistent concerns about safety and perceived risk. Encouraging young adults to initiate conversations with medical professionals about HPV vaccination may further strengthen engagement and informed decision-making. As parents could play a significant role in the vaccination process [33,43,45], empowering parents through education and provider outreach may further promote their active participation in vaccination decisions and support higher uptake.

Lastly, co-administering HPV with other adolescent vaccines through same-day or “bundled” vaccination approaches has been shown to significantly improve initiation and completion rates, reducing missed opportunities and offering a scalable strategy to strengthen routine adolescent immunization [78,79,80,81,82]. Recent advances in HPV-containing combination vaccines highlight a promising strategy to enhance coverage and timeliness; however, their applicability remains limited to future implementation beyond the scope of current delivery systems in the U.S. Midwest [83,84]. Although the World Health Organization and other countries have adopted single-dose HPV vaccination schedules, the United States has not yet approved a single-dose regimen; therefore, our findings based on 2–3-dose endpoints remain directly relevant to the current U.S. vaccination policy and reporting framework. Nonetheless, future adoption of a single-dose schedule could help address persistent barriers to vaccine initiation and series completion in the U.S. [85].

These study findings provide a guide to address barriers and promote facilitators through population-based interventions to enhance vaccination uptake among adolescents and young adults. Our study identified individual, interpersonal, and organizational determinants of vaccination uptake. Concerns about HPV vaccine side effects and a lack of clinician recommendations to get vaccinated limited both initiation and completion, whereas HPV awareness and knowing someone with cervical cancer facilitated both HPV vaccine initiation and completion. Belief in vaccine effectiveness and family support were key facilitators of vaccine completion. Additionally, beliefs in vaccine safety, healthcare coverage, and beliefs about HPV-related cancers’ seriousness also played a facilitator role in vaccine initiation.

Our cross-sectional survey of adolescents and young adults across the Midwest revealed low HPV vaccination uptake and highlighted multilevel, race- and ethnic-specific barriers, including concerns about side effects and the absence of provider recommendations, as well as key facilitators such as vaccine awareness, confidence in its effectiveness and safety, family support, and knowing someone with cervical cancer. These results emphasize the importance of multilevel strategies that combine consistent provider recommendations with family-centered engagement, clear communication about vaccine safety and efficacy, and expanded access and coverage. Given the self-reported vaccination data, cross-sectional design, and limited sample sizes for racial/ethnic groups, future longitudinal and intervention studies are needed to evaluate targeted approaches and reduce inequities. Advancing such evidence-based, multilevel interventions could improve vaccine initiation and completion rates, supporting progress toward national HPV coverage goals.

## Figures and Tables

**Table 1 vaccines-13-01175-t001:** Sociodemographic characteristics of sampled teenagers and young adults from the Midwest region in the US.

		Vaccination Status				
	Total	Fully	Initiated	Unvaccinated	Unknown	
		*n* (%)	*n* (%)	*n* (%)	*n* (%)	*p*-Value
	*n* = 1306	397 (30.4)	124 (9.5)	324 (24.8)	461 (35.3)	
Age Mean (SD)	18.9 (3.2)	19.5 (3.23)	19.4 (3.25)	19.0 (3.22)	18.0 (2.91)	<0.001
Sex						0.100
Male	570 (43.9)	159 (40.4)	50 (40.7)	159 (49.2)	202 (44.0)	
Female	729 (56.1)	235 (59.6)	73 (59.3)	164 (50.8)	257 (56.0)	
Race						0.021
NH White	682 (52.2)	229 (57.7)	61 (49.2)	141 (43.5)	251(54.5)	
NH Black or African American	272 (20.8)	71 (17.9)	30 (24.2)	83 (25.6)	88 (19.1)	
Hispanic or Latino	223 (17.1)	65 (16.4)	17 (13.7)	63 (19.4)	78 (16.9)	
Other *	129 (9.9)	32 (8.1)	16 (12.9)	37 (11.4)	44 (9.5)	
Education level						<0.001
Less than high school	519 (39.7)	120 (30.2)	37 (29.8)	124 (38.3)	238 (51.6)	
High school graduate or GED	439 (33.6)	129 (32.5)	52 (41.9)	117 (36.1)	141 (30.6)	
Some college or higher	348 (26.7)	148 (37.3)	35 (28.2)	83 (25.6)	82 (17.8)	
State						0.649
Illinois	259 (19.8)	81 (31.3)	26 (10.0)	58 (22.4)	94 (36.3)	
Ohio	243 (18.6)	89 (36.6)	26 (10.7)	64 (26.3)	64 (26.3)	
Michigan	187 (14.3)	55 (29.4)	25 (13.4)	41 (21.9)	66 (35.3)	
Indiana	126 (9.7)	31 (24.6)	11 (8.7)	33 (26.2)	51 (40.5)	
Missouri	117 (8.9)	31 (26.5)	7 (6.0)	35 (29.9)	44 (37.6)	
Wisconsin	97 (7.4)	29 (30.0)	7(7.2)	25 (25.8)	36 (37.1)	
Minnesota	91 (6.7)	31 (34.1)	6 (6.6)	25 (27.5)	29 (31.9)	
Kansas	58 (4.44)	17 (29.3)	4 (6.9)	12 (20.7)	25 (43.1)	
Iowa	53 (4.1)	10 (18.9)	5 (9.4)	13 (24.5)	25 (47.2)	
Nebraska	42 (3.22)	14 (33.3)	4 (9.5)	10 (23.8)	14 (33.3)	
South Dakota	17 (1.3)	5 (29.4)	1 (5.9)	4 (23.5)	7 (41.2)	
North Dakota	16 (1.23)	4 (25.0)	2 (12.5)	4 (25.0)	6 (37.5)	

Fully vaccinated = 2 or 3 doses, Initiated vaccination = 1 dose. NH = Non-Hispanic. * American Indian or Alaska native, Asian, Native Hawaiian or Pacific Islander and Other. Age differed across vaccination groups (Kruskal–Wallis, *p* < 0.0001).

**Table 2 vaccines-13-01175-t002:** Predictors of HPV vaccination stratified by HPV vaccination status.

Predictor	Category		Vaccination Status				
		Total	Unvaccinated	Initiated	Fully	Unknown	*p*-Value
Have you ever heard about HPV infection?	Yes	915 (70.0)	202 (62.35)	91 (73.4)	326 (82.12)	296 (64.21)	<0.01
	No	391 (30.0)	122 (37.7)	33 (26.6)	71 (17.9)	165 (35.8)	
Have you ever heard about cervical, anal, penile, vaginal, vulva, or oropharynx cancer?	Yes	974 (74.6)	238 (73.5)	93(75.0)	334 (84.13)	309 (67.03)	<0.01
	No	332 (25.4)	86 (25.5)	31 (25.0)	63 (15.9)	152 (33.0)	
Have you ever heard about the HPV vaccine?	Yes	844 (64.6)	166 (51.23)	92 (74.2)	323 (81.4)	263 (57.05)	<0.01
	No	462 (35.4)	158 (48.8)	32 (25.8)	74 (18.7)	198 (43.0)	
HPV infection can cause cervical, oropharyngeal, vaginal, vulvar, penile, anal, and rectal cancers	True	1102 (84.4)	267 (82.4)	106 (85.5)	353 (88.9)	376 (81.6)	0.0182
	False	204 (15.6)	57 (17.6)	18 (14.5)	44 (11.1)	85 (18.4)	
HPV vaccine can prevent cervical, oropharyngeal, vaginal, vulvar, penile, anal, and rectal cancers	True	942 (72.1)	230 (71.0)	84 (67.7)	306 (77.1)	322 (69.9)	0.0595
	False	364 (27.9)	94 (29.0)	40 (32.3)	91 (22.9)	139 (30.2)	
I am concerned about vaccine side effects	Disagree	375 (28.7)	77 (23.8)	40 (32.3)	146 (36.8)	112 (24.3)	<0.0001
	Neutral	538 (41.2)	126 (38.9)	45 (36.3)	161 (40.6)	206 (44.7)	
	Agree	393 (30.1)	121 (37.40	39 (31.5)	90 (22.7)	143 (31.0)	
It is better to get the disease and get protected naturally	Disagree	718 (55.0)	147 (45.4)	70 (56.5)	257 (64.7)	244 (52.9)	<0.0001
	Neutral	381 (29.2)	108 (33.3)	34 (2704)	87 (21.9)	152 (33.0)	
	Agree	207 (15.8)	69 (21.3)	20 (16.1)	53 (13.4)	65 (14.1)	
There are too many vaccines available to take	Disagree	508 (38.9)	98 (30.2)	50 (40.3)	184 (46.4)	176 (38.2)	0.0001
	Neutral	523 (40.0)	137 (42.3)	44 (35.5)	140 (35.3)	202 (43.8)	
	Agree	275 (21.1)	89 (27.5)	30 (24.2)	73 (18.4)	83 (18.0)	
I have a negative experience with vaccination	Disagree	710 (54.4)	153 (47.2)	64 (51.6)	240 (60.5)	253 (54.9)	<0.0001
	Neutral	373 (8.6)	99 (30.6)	35 (28.2)	85 (21.4)	154 (33.4)	
	Agree	223 (17.1)	72 (22.2)	25 (20.2)	72 (18.1)	54 (11.7)	
Vaccines are effective in preventing disease	Disagree	166 (12.7)	61 (18.8)	24 (19.3	40 (10.1)	41 (8.9)	<0.0001
	Neutral	378 (28.9)	117 (36.1)	31 (25.0)	86 (21.7)	144 (31.2)	
	Agree	762 (58.4)	146 (45.1)	69 (55.7)	271 (68.3)	276 (59.9)	
It is very important that I receive all the necessary routine vaccines	Disagree	205 (15.7)	81 (25.0)	23 (18.6)	42 (10.6)	59 (12.8)	<0.0001
	Neutral	400 (30.6)	112 (34.6)	35 (28.2)	82 (20.7)	171 (37.1)	
	Agree	701 (53.7)	131 (40.4)	66 (53.2)	273 (68.8)	231 (50.1)	
Vaccine is one way that I can ensure good health	Disagree	220 (16.8)	85 (26.3)	22 (17.7)	45 (11.3)	68 (14.8)	<0.0001
	Neutral	400 (30.6)	108 (33.3)	39 (31.4)	111 (28.0)	142 (30.8)	
	Agree	686 (52.5)	131 (40.3)	63 (50.8)	241 (60.7)	251 (54.5)	
I have a responsibility to get vaccinated for the protection of others	Disagree	191 (14.6)	74 (22.8)	22 (17.7)	40 (10.1)	55 (11.9)	<0.0001
	Neutral	408 (31.2)	118 (36.4)	32 (25.8)	99 (24.9)	159 (34.5)	
	Agree	707 (54.1)	132 (40.7)	70 (56.4)	258 (65.0)	247 (53.6)	
My doctor/healthcare provider recommended me to receive HPV vaccine	Disagree	334 (25.6)	148 (45.7)	28 (22.6)	38 (9.6)	120 (26.3)	<0.0001
	Neutral	493 (37.7)	105 (32.4)	41 (33.1)	98 (24.7)	249 (54.0)	
	Agree	479 (36.7)	71 (21.9)	55 (44.3)	261 (65.7)	92 (20.0)	
My family member recommends/supports me to receive HPV vaccine	Disagree	354 (27.1)	159 (49.1)	36 (29.0)	42 (10.6)	117 (25.4)	<0.0001
	Neutral	476 (36.5)	105 (32.41)	32 (25.8)	97 (24.4)	242 (52.5)	
	Agree	476 (36.4)	60 (18.5)	56 (45.2)	258 (65.0)	102 (22.1)	
I believe that HPV vaccine is beneficial to my health	Disagree	200 (15.3)	93 (28.7)	21 (16.9)	29 (7.3)	57 (12.4)	<0.0001
	Neutral	498 (38.1)	145 (44.8)	38 (30.7)	96 (24.2)	219 (47.5)	
	Agree	608 (46.6)	86 (26.5)	65 (52.4)	272 (68.5)	185 (40.1)	
I believe that HPV vaccine is safe	Disagree	180 (13.8)	70 (21.6)	29 (23.4)	36 (9.1)	45 (9.8)	<0.0001
	Neutral	513 (39.3)	156 (48.1)	31 (25.0)	93 (23.4)	233 (50.5)	
	Agree	613 (46.9)	98 (30.2)	64 (51.6)	268 (67.5)	183 (39.7)	
I believe that HPV vaccine is effective	Disagree	157 (12.0)	65 (20.1)	19 (15.3)	27 (6.8)	46 (10.0)	<0.0001
	Neutral	517 (39.6)	157 (48.5)	47 (37.9)	87 (21.9)	226 (49.0)	
	Agree	632 (48.4)	102 (31.5)	58 (46.8)	283 (71.3)	189 (41.0)	
I believe that I will become sexually active	Disagree	268 (20.5)	80 (24.7)	32 (25.8)	64 (16.1)	92 (20.0)	<0.0001
	Neutral	416 (31.9)	108 (33.3)	33 (26.6)	101 (25.4)	174 (37.7)	
	Agree	622 (47.6	136 (42.0)	59 (47.6)	232 (58.4)	195 (42.3)	
I believe that HPV infection can cause cervical cancer	Disagree	191 (14.6)	56 (17.3)	24 (19.4)	47 (11.8)	64 (13.9)	<0.0001
	Neutral	524 (40.1)	140 (43.2)	41 (33.1)	120 (30.2)	223 (48.4)	
	Agree	591 (45.3)	128 (39.5)	59 (47.6)	230 (57.9)	174 (37.7)	
I believe that HPV vaccine will prevent cervical, oropharyngeal, vaginal, vulvar, penile, anal, and rectal cancers for self and others	Disagree	165 (12.6)	55 (17.0)	23 (18.6)	38 (9.6)	49 (10.6)	<0.0001
	Neutral	534 (40.9)	149 (46.0)	36 (29.0)	115 (29.0)	234 (50.8)	
	Agree	607 (46.5)	120 (37.0)	65 (52.4)	244 (61.5)	178 (38.6)	
HPV vaccine is too expensive	Disagree	351 (26.9)	92 (28.4)	36 (29.0)	140 (35.3)	83 (18.0)	<0.0001
	Neutral	706 (54.0)	175 (54.0)	52 (42.0)	178 (44.8)	301 (65.3)	
	Agree	249 (19.1)	57 (17.6)	36 (29.0)	79 (19.9)	77 (16.7)	
My religious belief goes against me getting HPV vaccine	Disagree	769 (58.9)	177 (54.6)	69 (55.7)	264 (66.5)	259 (56.2)	<0.0001
	Neutral	358 (27.4)	95 (29.3)	25 (20.2)	76 (19.1)	162 (35.1)	
	Agree	179 (13.7)	52 (16.1)	30 (24.2)	57 (14.4)	40 (8.7)	
Objections in getting HPV vaccine from religious authorities will prevent me from getting the vaccine	Disagree	670 (51.3)	144 (44.4)	68 (54.8)	235 (59.2)	223 (48.4)	<0.0001
	Neutral	453 (34.7)	131 (40.4)	34 (27.4)	98 (24.7)	190 (41.2)	
	Agree	183 (14.0)	49 (15.1)	22 (17.7)	64 (16.1)	48 (10.4)	
Friends who disapprove of getting HPV vaccine will prevent me from getting the vaccine	Disagree	705 (54.0)	151 (46.6)	68 (54.8)	247 (62.2)	239 (51.8)	<0.0001
	Neutral	417 (31.9)	126 (38.9)	30 (24.2)	87 (21.9)	174 (37.7)	
	Agree	184 (14.1)	47 (14.5)	26 (21.0)	63 (15.9)	48 (10.4)	
I am afraid that HPV vaccine injection may cause pain	Disagree	477 (36.5)	119 (36.7)	44 (35.5)	180 (45.3)	134 (29.0)	<0.0001
	Neutral	489 (37.5)	121 (37.4)	35 (28.2)	121(30.5)	212 (46.0)	
	Agree	340 (26.0)	84 (25.9)	45 (36.3)	96 (24.2)	115 (25.0)	
I will not have time to get HPV vaccine	Disagree	572 (43.8)	116 (35.8)	51 (41.1)	230 (57.9)	175 (38.0)	<0.0001
	Neutral	504 (38.6)	133 (41.0)	41 (33.1)	112 (28.2)	218 (47.3)	
	Agree	230 (17.6)	75 (23.2)	32 (25.8)	55 (13.9)	68 (14.7)	
I do not know how to make an appointment to get HPV vaccine	Disagree	538 (41.2)	123 (38.0)	56 (45.2)	219 (55.2)	140 (30.4)	<0.0001
	Neutral	462 (35.4)	118 36.4)	37 (29.8)	97 (24.4)	210 (45.6)	
	Agree	306 (23.4)	83 (25.6)	31 (25.0)	81 (20.4)	111 (24.1)	
My health insurance does not cover the HPV vaccine	Disagree	447 (34.2)	99 (30.6)	49 (39.5)	193 (48.6)	106 (23.0)	<0.01
	Neutral	661 (50.6)	171 (52.8)	41 (33.1)	144 (36.3)	305 (66.2)	
	Agree	198 (15.2)	54 (16.7)	34 (27.4)	60 (15.1)	50 (10.8)	
I am too young for getting vaccination	Disagree	584 (44.7)	137 (42.3)	64 (51.6)	242 (61.0)	141 (30.6)	<0.01
	Neutral	486 (37.2)	113 (34.9)	34 (27.4)	108 (27.2)	231 (50.1)	
	Agree	236 (18.1)	74 (22.8)	26 (21.0)	47 (11.8)	89 (19.3)	
I believe I am at risk of getting HPV	Disagree	721 (55.2)	199 (61.4)	62 (50.0)	211 (53.1)	249 (54.0)	0.0051
	Neutral	421 (32.2)	89 (27.5)	44 (35.5)	120 (30.2)	168 (36.4)	
	Agree	164 (12.6)	36 (11.1)	18 (14.5)	66 (16.6)	44 (9.5)	
I will likely contract HPV infection	Disagree	801 (61.3)	207 (63.9)	69 (55.7)	258 (65.0)	267 (57.9)	0.0048
	Neutral	389 (29.8)	92 (28.4)	37 (29.8)	98 (24.7)	162 (35.1)	
	Agree	116 (8.9)	25 (7.7)	18 (14.5)	41 (10.3)	32 (6.9)	
I am at risk of getting cervical, oropharyngeal, vaginal, penile, anal, or rectal cancer	Disagree	782 (59.9)	206 (63.6)	75 (60.5)	235 (59.2)	266 (57.5)	0.0065
	Neutral	393 (30.1)	93 (28.7)	31 (25.0)	109 (27.5)	160 (34.7)	
	Agree	131 (10.0)	25 (7.7)	18 (14.5)	53 (13.4)	35 (4.6)	
I will likely contract cervical, oropharyngeal, vaginal, penile, or rectal cancer	Disagree	839 (64.2)	214 (66.0)	73 (58.8)	263 (66.3)	289 (62.7)	0.0002
	Neutral	352 (27.0)	87 (26.9)	30 (24.2)	89 (22.4)	146 (31.7)	
	Agree	115 (8.8)	23 (7.1)	21 (17.0)	45 (11.3)	26 (5.6)	
I believe that HPV infection is severe	Disagree	193 (14.8)	53 (16.4)	24 (19.3)	49 (12.3)	67 (14.5)	<0.001
	Neutral	422 (32.3)	116 (35.8)	27(21.8)	98 (24.7)	181 (39.3)	
	Agree	691 (52.9)	155 (47.8)	73 (58.9)	250 (63.0)	213 (46.2)	
I believe that HPV infection is serious	Disagree	159 (12.2)	51 (15.7)	19 (15.3)	32 (8.6)	55 (11.9)	<0.001
	Neutral	322 (24.7)	93 (28.7)	25 (20.2)	71 (17.9)	133 (28.9)	
	Agree	825 (63.2)	180 (55.6)	80 (64.5)	292 (73.5)	273 (59.2)	
I believe that cervical, oropharyngeal, vaginal, vulvar, penile, anal, and rectal cancers are severe	Disagree	180 (13.8)	48 (14.8)	23 (18.6)	51 (18.9)	58 (12.6)	0.0012
	Neutral	342 (26.2)	104 (32.1)	26 (21.0)	79 (19.9)	133 (28.9)	
	Agree	784 (60.0)	172 (53.1)	75 (60.5)	267 (67.3)	270 (58.6)	
I believe that cervical, oropharyngeal, vaginal, vulvar, penile, anal, and rectal cancers are serious	Disagree	169 (12.9)	50 (15.4)	28 (22.6)	39 (9.8)	52 (11.3)	<0.001
	Neutral	303 (23.2)	94 (29.0)	16 (12.9)	70 (17.6)	123 (26.7)	
	Agree	834 (63.9)	180 (55.6)	80 (64.5)	288 (73.6)	286 (63.0)	
If I knew a woman with cervical cancer, I would be motivated to get the HPV vaccine	Yes	710 (54.4)	136 (42.0)	85 (68.6)	290 (73.05)	199 (43.2)	<0.001
	No, not sure	596 (45.6)	188 (58.02)	39 (31.45)	107 (27.0)	262 (56.83)	
If I knew a family member who has developed HPV infection related to cancer, I would be motivated to get the HPV vaccine	Yes	755 (57.8)	151 (46.6)	75 (60.5)	299 (75.3)	230 (49.9)	<0.001
	No, not sure	551 (42.2)	173 (53.4)	49 (39.5)	98 (24.7)	231 (50.1)	
I believe I can succeed/achieve getting the HPV vaccine even when things are tough	Disagree	186 (14.2)	67 (20.7)	26 (21.0)	42 (10.6)	51 (11.1)	<0.001
	Neutral	519 (39.7)	145 (44.8)	42 (33.9)	107 (27.0)	255 (48.8)	
	Agree	601 (46.0)	112 (34.6)	56 (45.2)	248 (62.5)	185 (40.1)	
I am confident that I can get the HPV vaccine overcoming challenges	Disagree	176 (13.5)	73 (22.5)	21 (16.9)	33 (8.3)	49 (10.6)	<0.001
	Neutral	525 (40.2)	142 (43.8)	42 (33.9)	115 (29.0)	226 (49.0)	
	Agree	605 (46.3)	109 (33.6)	61 (49.2)	249 (62.7)	186 (40.3)	
I believe society expects me to get the HPV vaccine	Disagree	259 (19.8)	89 (27.5)	27 (21.8)	56 (14.1)	87 (18.9)	<0.001
	Neutral	586 (44.9)	148 (45.7)	55 (44.3)	145 (36.5)	238 (51.6)	
	Agree	461 (35.3)	87 (26.9)	42 (33.9)	196 (49.4)	136 (29.5)	
I have a responsibility to get the HPV vaccine for the protection of others	Disagree	219 (16.8)	83 (25.6)	25 (20.2)	46 (11.6)	65 (14.1)	<0.001
	Neutral	490 (37.5)	128 (39.5)	38 (30.7)	114 (28.7)	210 (45.6)	
	Agree	597 (45.7)	113 (34.9)	61 (40.2)	237 (59.7)	186 (40.3)	

**Table 3 vaccines-13-01175-t003:** Multinomial logistic regression model for predicting HPV vaccination uptake among teenagers and young adults in the Midwest region of the US.

Predictor	Category	Vaccination Status vs. Unvaccinated	Odds Ratio Estimates	95% Confidence Intervals	
Have you ever heard about cervical, anal, penile, vaginal, vulva, or oropharynx cancer?	Yes vs. No	Initiated vaccination	0.64	0.37	1.09
		Fully vaccinated	0.92	0.59	1.44
Have you ever heard about the HPV vaccine? (awareness)	Yes vs. No	Initiated vaccination	2.43	1.46	4.04
		Fully vaccinated	2.39	1.61	3.55
I am concerned about vaccine side effects	Disagree vs. Neutral	Initiated vaccination	1.05	0.59	1.84
		Fully vaccinated	1.05	0.67	1.62
	Agree vs. Neutral	Initiated vaccination	0.78	0.45	1.35
		Fully vaccinated	0.52	0.34	0.80
Vaccine is one way that I can ensure good health	Disagree vs. Neutral	Initiated vaccination	0.63	0.32	1.26
		Fully vaccinated	0.74	0.43	1.29
	Agree vs. Neutral	Initiated vaccination	0.74	0.42	1.29
		Fully vaccinated	0.71	0.46	1.11
My doctor/healthcare provider recommended me to receive HPV vaccine	Disagree vs. Neutral	Initiated vaccination	0.47	0.24	0.90
		Fully vaccinated	0.46	0.27	0.80
	Agree vs. Neutral	Initiated vaccination	1.04	0.55	1.95
		Fully vaccinated	1.58	0.97	2.58
My family member recommends/supports me to receive HPV vaccine	Disagree vs. Neutral	Initiated vaccination	0.91	0.47	1.77
		Fully vaccinated	0.46	0.27	0.79
	Agree vs. Neutral	Initiated vaccination	1.87	0.99	3.54
		Fully vaccinated	2.32	1.42	3.77
I believe that HPV vaccine is safe	Disagree vs. Neutral	Initiated vaccination	2.87	1.36	6.06
		Fully vaccinated	1.75	0.92	3.33
	Agree vs. Neutral	Initiated vaccination	2.01	1.04	3.88
		Fully vaccinated	1.43	0.87	2.37
I believe that HPV vaccine is effective	Disagree vs. Neutral	Initiated vaccination	0.52	0.24	1.15
		Fully vaccinated	0.84	0.42	1.68
	Agree vs. Neutral	Initiated vaccination	0.63	0.34	1.17
		Fully vaccinated	1.77	1.08	2.91
My health insurance does not cover the HPV vaccine	Disagree vs. Neutral	Initiated vaccination	1.27	0.73	2.23
		Fully vaccinated	1.27	0.82	1.95
	Agree vs. Neutral	Initiated vaccination	1.86	1.00	3.45
		Fully vaccinated	1.17	0.70	1.96
I am too young for getting vaccination	Disagree vs. Neutral	Initiated vaccination	1.20	0.67	2.14
		Fully vaccinated	1.11	0.71	1.73
	Agree vs. Neutral	Initiated vaccination	1.06	0.55	2.06
		Fully vaccinated	0.64	0.37	1.11
I believe that cervical, oropharyngeal, vaginal, vulvar, penile, anal, and rectal cancers are serious	Disagree vs. Neutral	Initiated vaccination	3.63	1.61	8.19
		Fully vaccinated	1.16	0.61	2.22
	Agree vs. Neutral	Initiated vaccination	1.95	0.99	3.87
		Fully vaccinated	0.95	0.59	1.52
If I knew a woman with cervical cancer, I would be motivated to get the HPV vaccine	Yes vs. No	Initiated vaccination	2.35	1.42	3.87
		Fully vaccinated	1.81	1.24	2.65

## Data Availability

The data presented in this study are available on request from the corresponding author. The data are not publicly available due to privacy/ethical issues.

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
