# Peer review of "HPV Vaccination in the U.S. Midwest: Barriers and Facilitators of Initiation and Completion in Adolescents and Young Adults"

_vaccines, 2025, doi:10.3390/vaccines13111175_

Round 1
Reviewer 1 Report
Comments and Suggestions for Authors
The topic addressed is of significant public health relevance, particularly in the current global context of declining vaccination coverage, particularly among adolescents and young adults. The study investigates the key determinants influencing HPV vaccination initiation and completion in a population in the Midwestern United States. The research question is timely and relevant, with potentially valuable descriptive findings that could significantly contribute to the scientific understanding of HPV vaccine uptake. However, significant revisions are needed to improve the clarity, methodological rigor, and interpretive depth of the work before it can be considered for publication.
The following comments and recommendations are intended to assist the authors in strengthening their manuscript. My suggestions are therefore presented in detail, point by point, as described below.
Title: The title is overly long and could be made more concise to improve clarity and focus.
Introduction: The research objectives are clearly stated; however, the underlying hypotheses require further elaboration to better articulate the theoretical and contextual framework guiding the study. Notably, the issue of vaccine hesitancy and its determinants within the study cohort are not discussed in the Introduction and are only briefly mentioned in the Discussion (around line 335). In my view, this represents a significant gap in the conceptual framing of the paper. Understanding the determinants of vaccine hesitancy is now essential for designing effective, evidence-based strategies to improve vaccine uptake and adherence (see doi: 10.3390/ijerph19074359). I therefore encourage the authors to explore this theme in greater depth, drawing on recent evidence (doi: 10.1016/j.vaccine.2025.127401; doi: 10.1038/s41598-025-94067-1). Addressing this aspect would add conceptual depth and enhance the scientific relevance of the study.
A more critical examination of the psychological and socio-cultural dimensions emerging from the questionnaire would substantially strengthen the background.
Lines 30–43: The Introduction outlines the public health importance of HPV vaccination, but lacks an updated contextualization of current vaccination trends among adolescents and young adults, both nationally and globally (data for 2022–2023). While this may reflect the evidence available at the time the survey was administered, including more recent WHO or CDC data would provide a more current and realistic picture.
- Lines 41–58: The discussion of factors influencing HPV vaccine adherence remains superficial. The Introduction would benefit from a more comprehensive synthesis of existing evidence on the psychosocial, cultural, and structural determinants of vaccination initiation and completion, particularly within the conceptual framework of vaccine hesitancy, as defined by the WHO SAGE Working Group. In this context, a brief consideration of the digital environment and its role in amplifying misinformation would also be helpful.
- The cited references (Refs. 18, 25, 73) are relevant, however, the inclusion of a commentary addressing HPV vaccine hesitancy in the post-COVID-19 era is recommended to reflect recent changes in public confidence in immunization programs.
- Lines 74–79: The transition from context to study objectives could be clearer. I suggest revising the final paragraph of the Introduction to explicitly state the study's objectives, hypotheses, and intended implications for public health practice.
Methods
The survey was conducted among adolescents and young adults in the Midwestern United States. A brief description of the study context is needed to assess how closely the sample reflects the target population.
- In line 87, participants were recruited through the Qualtrics research panel. The recruitment process is not described in sufficient detail. The statement "details are reported elsewhere" is inadequate; essential methodological information should be provided within the manuscript or its supplementary materials. Authors should clarify how panel members were selected and whether the sample was stratified or weighted to represent the Midwestern population of adolescents and young adults.
- Using an online panel may introduce self-selection bias and overrepresentation of certain sociodemographic groups, such as those with higher education or digital access. Authors should explain how these potential biases were addressed.
- Ethical procedures also require clarification. The manuscript must specify how the consent of minors was obtained and whether parental consent was required. If parental consent was withdrawn, the reason must be explicitly stated and reference made to the protocol and how this is compatible with payment for a minor's participation.
Results
There are several inconsistencies in the presentation of the results that need to be clarified. The abstract and main text state that 1,309 adolescents and young adults were interviewed, while Table 1 reports a total of 1,306. This discrepancy needs to be explained. The percentages in Table 1 also differ from the text (e.g., 30.7% for 397 respondents). Minor rounding discrepancies are acceptable, but all numerators, denominators, and percentages should be checked and harmonized throughout the manuscript.
The manuscript does not indicate how many responses were incomplete or excluded from the final analysis. Providing this information, ideally in a brief flowchart showing the number of subjects invited, screened, eligible, included, and excluded, would improve transparency. It is unclear whether sensitivity analyses were conducted to test the robustness of the results or whether corrections for multiple testing were applied. These details are important for assessing the reliability of the reported associations. Finally, please review the formatting of statistical values. Some typographical inconsistencies were noted, such as the use of "OR –" instead of "OR (95% CI – )" or "OR;". Ensuring consistency between tables and text would improve readability and professionalism.
Discussion: The discussion is well-presented, and the results have been described by exploring different interpretations, with extensive supporting literature.
However, it is suggested that clarification be given to how the research conducted can contribute to theoretical and empirical progress in combating vaccine hesitancy in the indicated cohort (I suggest consulting the 3C or 5C model proposed by the WHO). Furthermore, consideration of the digital divide in socially disadvantaged communities in today's information-abundant environment has been omitted, despite this being one of the additional determinants negatively impacting vaccine adherence. Similarly, the authors refer to misinformation but do not discuss the impact of this determinant, which is known to be a serious obstacle to vaccination efforts; this aspect could be further explored with extensive supporting literature, as it represents one of the main factors of vaccine hesitancy among adolescents, especially regarding HPV.
Author Response
The topic addressed is of significant public health relevance, particularly in the current global context of declining vaccination coverage, particularly among adolescents and young adults. The study investigates the key determinants influencing HPV vaccination initiation and completion in a population in the Midwestern United States. The research question is timely and relevant, with potentially valuable descriptive findings that could significantly contribute to the scientific understanding of HPV vaccine uptake. However, significant revisions are needed to improve the clarity, methodological rigor, and interpretive depth of the work before it can be considered for publication.
- The following comments and recommendations are intended to assist the authors in strengthening their manuscript. My suggestions are therefore presented in detail, point by point, as described below.
Title: The title is overly long and could be made more concise to improve clarity and focus.
Response: We shortened the title to improve clarity and focus. The revised title reads “HPV Vaccination in the U.S. Midwest: Barriers and Facilitators of Initiation and Completion in Adolescents and Young Adults”. Please see lines 2 and 3.
- Introduction: The research objectives are clearly stated; however, the underlying hypotheses require further elaboration to better articulate the theoretical and contextual framework guiding the study. Notably, the issue of vaccine hesitancy and its determinants within the study cohort are not discussed in the Introduction and are only briefly mentioned in the Discussion (around line 335). In my view, this represents a significant gap in the conceptual framing of the paper. Understanding the determinants of vaccine hesitancy is now essential for designing effective, evidence-based strategies to improve vaccine uptake and adherence (see doi: 10.3390/ijerph19074359). I therefore encourage the authors to explore this theme in greater depth, drawing on recent evidence (doi:0.1016/j.vaccine.2025.127401; doi: 10.1038/s41598-025-94067-1). Addressing this aspect would add conceptual depth and enhance the scientific relevance of the study. A more critical examination of the psychological and socio-cultural dimensions emerging from the questionnaire would substantially strengthen the background.
Response: Thank you for the comments. In addition to the literature review we summarized regarding the factors that could affect vaccine uptake, “Studies have identified factors such as geographic, racial, gender, age and race disparities as reasons for the decreased uptake of the HPV vaccine in the US. Individual factors such as attitudes, perceived social expectations to get vaccinated, beliefs of self-ability to accomplish vaccination, and perceived risks of HPV-related disease have been shown to influence vaccination intention. Other reported barriers include lack of knowledge about HPV and the HPV vaccine, concerns regarding its safety and efficacy, financial constraints, and fears of discrimination associated with receiving the vaccine.”, we have expanded the introduction to include determinants of vaccine hesitancy based oh the theoretical frameworks. The added text reads: “ Additionally, newer studies have found areal,societal and policy level factors influencing vaccination uptake.” (Line 58-49) “Moreover, psychosocial drivers (i.e. confidence in vaccine safety, perceived disease risk, social norms, and trust in providers) influence initiation of HPV vaccination, while cultural beliefs and historical mistrust may affect confidence across communities. Structural barriers such as access, cost, and continuity of care also influence completion of HPV vaccination, emphasizing that both psychological trust and systemic convenience are necessary for sustained vaccination coverage.” (Lines 95-100).
- Lines 30–43: The Introduction outlines the public health importance of HPV vaccination but lacks an updated contextualization of current vaccination trends among adolescents and young adults, both nationally and globally (data for 2022–2023). While this may reflect the evidence available at the time the survey was administered, including more recent WHO or CDC data would provide a more current and realistic picture.
Response: We have updated the HPV vaccination data for the adolescents based on the latest estimate published in CDC in 2025. (e.g., MMWR Morb Mortal Wkly Rep 2025;74:466–472.DOI: http://dx.doi.org/10.15585/mmwr.mm7430a1). For young adults the latest data we found was from 2022. The updated text reads: “Coverage among adolescents aged 13-17 years was 62.9% in 2024, and among young adults 47% in 2022.”
- Lines 41–58: The discussion of factors influencing HPV vaccine adherence remains superficial. The Introduction would benefit from a more comprehensive synthesis of existing evidence on the psychosocial, cultural, and structural determinants of vaccination initiation and completion, particularly within the conceptual framework of vaccine hesitancy, as defined by the WHO SAGE Working Group. In this context, a brief consideration of the digital environment and its role in amplifying misinformation would also be helpful.
Response: We appreciate the reviewer’s insights we have highlighted the importance of psychosocial, cultural, and structural determinants of HPV vaccination, as well as the role of misinformation and the digital environment in our introduction. The added text reads” Moreover, psychosocial, drivers (i.e. confidence in vaccine safety, perceived disease risk, social norms, and trust in providers) primarily influence initiation of HPV vaccination, while cultural beliefs and historical mistrust may affect confidence across communities. Structural barriers such as access, cost, and continuity of care influence completion, emphasizing that both psychological trust and systemic convenience are necessary for sustained vaccination coverage. Furthermore, digital environment may play a role in vaccine uptake by increasing misinformation about the HPV, HPV related cancer and the vaccine among the target population.” Please see line 94-101.
- The cited references (Refs. 18, 25, 73) are relevant, however, the inclusion of a commentary addressing HPV vaccine hesitancy in the post-COVID-19 era is recommended to reflect recent changes in public confidence in immunization programs.
Response: We have commented the issue of vaccine hesitancy in the post COVID era in the Introduction that reads “
- Lines 74–79: The transition from context to study objectives could be clearer. I suggest revising the final paragraph of the Introduction to explicitly state the study's objectives, hypotheses, and intended implications for public health practice.
Response: We have revised the last paragraph in the background. The study aims followed by the hypothesis and the implication. Please see lines 102-109.
The revised text reads “The aim of this study was to identify facilitators and barriers to HPV vaccination uptake among adolescents and young adults in the Midwest region of the US and to examine whether these facilitators and barriers differed across racial subgroups within this population. We hypothesized that attitudes, beliefs, and social influences, and would be key predictors of vaccination intentions and behaviors, and that these determinants might vary by race and cultural context. By identifying key barriers and facilitators the study seeks to inform evidence-based, culturally responsive strategies to improve HPV vaccination coverage and equity in public health practice”
Methods
- The survey was conducted among adolescents and young adults in the Midwestern United States. A brief description of the study context is needed to assess how closely the sample reflects the target population.
Response: We appreciate the reviewer’s comment. We have added a few lines in the methods that summarize the demographic characteristics of the target population. The added text reads “The Midwestern region had a combined population of approximately 69 million residents in 2023. According to American Community Survey (ACS) estimates, adolescents aged 13–17 years represent roughly 7 to 8% of the total population in the region and young adults aged 18 to 26 years represent 11–12%. Across states in the region, the population distributions among adolescents and young adults was close to parity. (Lines 117-126).
- In line 87, participants were recruited through the Qualtrics research panel. The recruitment process is not described in sufficient detail. The statement "details are reported elsewhere" is inadequate; essential methodological information should be provided within the manuscript or its supplementary materials. Authors should clarify how panel members were selected and whether the sample was stratified or weighted to represent the Midwestern population of adolescents and young adults.
Response: We have expanded the Methods section to provide additional details about participant recruitment procedure. The added text reads “(Lines 133-125, 140-142).
- Using an online panel may introduce self-selection bias and overrepresentation of certain sociodemographic groups, such as those with higher education or digital access. Authors should explain how these potential biases were addressed.
Response: We appreciate this important observation. In response, we have expanded the Methods to acknowledge the potential for self-selection bias when using online panels and describe the steps Qualtrics employs to mitigate it, including double–opt-in recruitment, demographic quota sampling, and quality control procedures. (Lines 144-149). The added text reads “ Although using an online panel may introduce self-selection bias, Qualtrics mitigates this risk through actively managed, double–opt-in recruitment, demographic quota sampling, and rigorous quality controls (e.g., attention checks, IP validation, and digital fingerprinting). Weighting procedures applied when necessary to align the sample with population benchmarks, thereby enhancing representativeness and data reliability.”
- Ethical procedures also require clarification. The manuscript must specify how the consent of minors was obtained and whether parental consent was required. If parental consent was withdrawn, the reason must be explicitly stated and reference made to the protocol and how this is compatible with payment for a minor's participation.
Response: We thank the reviewer for this important observation. The study was conducted through Qualtrics Panels, which adheres to institutional and federal ethical standards for online research. We have expanded the text regarding the assent/consent procedure for the participants. The added text reads “Participants provided assent by checking a box after reading a consent form outlining the study’s purpose, confidentiality, risks, and benefits. Only individuals who gave assent proceeded to complete the survey. Qualtrics provides proprietary incentives based on survey length, participant profile, and completion rate. Participants who answer at least 50% of questions receive compensation, which may include cash, points, gift cards, or other rewards.”
Because this survey posed a minimal risk to the respondents who were reached from existing pools of research panel lists of data processor company, the IRB waived parental or guardian consent/permission for teenagers
Regarding compensation, Qualtrics administers proprietary incentives directly to participants (or their guardians, in the case of minors) based on survey length and completion rate. The research team did not determine or distribute compensation amounts. We have clarified these procedures in the Methods – Ethical Considerations section. Please refer to lines: 189-194.
Results
- There are several inconsistencies in the presentation of the results that need to be clarified. The abstract and main text state that 1,309 adolescents and young adults were interviewed, while Table 1 reports a total of 1,306. This discrepancy needs to be explained. The percentages in Table 1 also differ from the text (e.g., 30.7% for 397 respondents). Minor rounding discrepancies are acceptable, but all numerators, denominators, and percentages should be checked and harmonized throughout the manuscript.
Response: We appreciate the reviewer’s careful attention to detail. We have edited the results for inconsistencies in numerators, denominators, and percentages. The discrepancy noted between the text (1,309 participants) and Table 1 (1,306 participants) was due to a typographical error in the main text. The correct number of respondents is 1,306, as reported in Table 1. We have corrected this throughout the manuscript and rechecked all numerators, denominators, and percentages to ensure internal consistency and accurate rounding in the abstract, results, and tables.
- The manuscript does not indicate how many responses were incomplete or excluded from the final analysis. Providing this information, ideally in a brief flowchart showing the number of subjects invited, screened, eligible, included, and excluded, would improve transparency. It is unclear whether sensitivity analyses were conducted to test the robustness of the results or whether corrections for multiple testing were applied. These details are important for assessing the reliability of the reported associations. Finally, please review the formatting of statistical values. Some typographical inconsistencies were noted, such as the use of "OR –" instead of "OR (95% CI – )" or "OR;". Ensuring consistency between tables and text would improve readability and professionalism.
Response: We thank the reviewer for these helpful suggestions. Qualtrics collected and provided data for 1306 individuals. All 1,306 survey responses were complete and included in the final analysis. However, we did not get information about the number of individuals eligible, contacted, because of this reason we have included it as part of our study limitations in our study. Please refer to lines: 387-388. we have also reviewed and standardized the formatting of all statistical values to ensure consistency between the tables and text (e.g., “OR (95% CI – )” format).
In addition, in order to reduce the chance of finding false significant results due to multiple comparisons, corrections were made deploying the Benjamini–Hochberg method.
The added text reads “. in order to reduce the chance of finding false significant results due to multiple comparisons, corrections were made deploying the Benjamini–Hochberg method.“(Lines 206-208)
Discussion
- The discussion is well-presented, and the results have been described by exploring different interpretations, with extensive supporting literature. However, it is suggested that clarification be given to how the research conducted can contribute to theoretical and empirical progress in combating vaccine hesitancy in the indicated cohort (I suggest consulting the 3C or 5C model proposed by the WHO). Furthermore, consideration of the digital divide in socially disadvantaged communities in today's information-abundant environment has been omitted, despite this being one of the additional determinants negatively impacting vaccine adherence. Similarly, the authors refer to misinformation but do not discuss the impact of this determinant, which is known to be a serious obstacle to vaccination efforts; this aspect could be further explored with extensive supporting literature, as it represents one of the main factors of vaccine hesitancy among adolescents, especially regarding HPV.
Response: We thank the reviewer for these insightful suggestions. In response, we have strengthened the Discussion to better situate our findings within established theoretical frameworks of vaccine hesitancy ,explicitly linking them to constructs from the WHO 3C/5C models. The revised text reads” Lastly, our findings align with and extend established behavioral models of vaccine hesitancy.58,76 Consistent with the World Health Organization 5C framework, concerns about vaccine safety and side effects may reflect low confidence, whereas awareness, belief in vaccine effectiveness, and family support may reflect high confidence and collective responsibility. Perceived seriousness of HPV-related cancers addresses complacency, and healthcare coverage reflects the constraints dimension. These findings contribute to theoretical progress by showing how these factors work together to influence whether adolescents and young adults start and complete the HPV vaccination.” (Lines 373-380)
Additionally, we have acknowledged as a limitation that our study did not assess broader contextual determinants, including socio-digital inequalities that may restrict access to credible vaccine information, and the influence of misinformation, which is increasingly recognized as a major driver of HPV vaccine hesitancy. These factors were outside the scope of our data but are important avenues for future research. Please refer to lines 396-399.
Reviewer 2 Report
Comments and Suggestions for Authors
This project will conduct a study on the HPV vaccination coverage rate and the factors that may affect the vaccination rate among teenagers and young adults in 12 states of the Midwestern region of the United States, and will compare different racial and ethnic groups. It will clarify the common promoting factors and hindering factors that affect HPV vaccination among various races and ethnic groups. The promoting factors include: understanding the safety and effectiveness of the vaccine itself; awareness of HPV susceptibility and HPV-related diseases and cancers (such as cervical cancer); information transmission, publicity, vaccination recommendations and family support from doctors and healthcare providers; social intervention measures - vaccine research institutions and manufacturers further improve the safety and effectiveness of the vaccine, increase medical coverage and expand publicity, and improve the accessibility of HPV vaccines through medical insurance. The hindering factors include: those from research and production aspects: side effects of HPV vaccines; those from personal and family aspects: insufficient awareness of HPV susceptibility and related diseases and cancers, the belief that one is too young to be vaccinated against HPV, and the lack of recognition that "vaccines are an effective way to ensure health". It has pointed out the key improvement directions and public health intervention measures - addressing multiple-level hindrances and fully leveraging the promoting factors, so that the HPV vaccination coverage rate among teenagers and young adults in the Midwestern region of the United States can reach the 80% target set in the "Healthy People 2030" plan, and provide reference and guidance for the increase in HPV vaccination coverage in other regions of the United States and other countries.
Areas needing improvement:
- In line 174, the figures 55-58% should be 55%-58%.
- The third table in Table 1 is missing the first line. The second line should extend to the left to cover all the content on the left side of the table.
- Table 2 lacks the topmost line of the table.
- Since all the authors belong to the same unit, there is no need to mark "1" in the upper right corner after the author names. The "1" on the left side of the author's unit in the 6th line should also be deleted.
Author Response
This project will conduct a study on the HPV vaccination coverage rate and the factors that may affect the vaccination rate among teenagers and young adults in 12 states of the Midwestern region of the United States, and will compare different racial and ethnic groups. It will clarify the common promoting factors and hindering factors that affect HPV vaccination among various races and ethnic groups. The promoting factors include: understanding the safety and effectiveness of the vaccine itself; awareness of HPV susceptibility and HPV-related diseases and cancers (such as cervical cancer); information transmission, publicity, vaccination recommendations and family support from doctors and healthcare providers; social intervention measures - vaccine research institutions and manufacturers further improve the safety and effectiveness of the vaccine, increase medical coverage and expand publicity, and improve the accessibility of HPV vaccines through medical insurance. The hindering factors include: those from research and production aspects: side effects of HPV vaccines; those from personal and family aspects: insufficient awareness of HPV susceptibility and related diseases and cancers, the belief that one is too young to be vaccinated against HPV, and the lack of recognition that "vaccines are an effective way to ensure health". It has pointed out the key improvement directions and public health intervention measures - addressing multiple-level hindrances and fully leveraging the promoting factors, so that the HPV vaccination coverage rate among teenagers and young adults in the Midwestern region of the United States can reach the 80% target set in the "Healthy People 2030" plan, and provide reference and guidance for the increase in HPV vaccination coverage in other regions of the United States and other countries.
Areas needing improvement:
- In line 174, the figures 55-58% should be 55%-58%.
Response: Thank you for catching this type, we have corrected it to 55%-58%.
- The third table in Table 1 is missing the first line. The second line should extend to the left to cover all the content on the left side of the table.
Response: We have reformatted Table 1 for easier understanding.
- Table 2 lacks the topmost line of the table.
Response: We have reformatted Table 1 for easier understanding.
- Since all the authors belong to the same unit, there is no need to mark "1" in the upper right corner after the author names. The "1" on the left side of the author's unit in the 6th line should also be deleted.
Response: We have edited the credentials to address this comment. Please see lines 5 and 6.
Reviewer 3 Report
Comments and Suggestions for Authors
Dear authors,
Your write-up provides a comprehensive test of predictors of HPV vaccine uptake with 44 predictor variables. It includes beliefs, attitudes, sociological variables such as friend and family support, and demographic factors. The sampling for the survey also has good representation of different racial and ethnic groups. Overall, the study is well designed and methodologically sound. It is NIH funded, so it has to be.
One obvious limitation of your survey data the large number of the respondents who were unsure of their vaccination status. Fortunately, the cross-tabulation tables in the supplementary material allow the observer to compare between the different groups across the independent predictor variables and it seems that the "vaccination unknown" group bears some resemblance to the "known unvaccinated group".
The results of this study provides some useful information for public health practitioners trying to develop strategies to increase vaccine uptake of the HPV vaccine. When I read a manuscript such as this, I am primarily interested in the public health practice implications of the research. In the current piece, I was primarily interested in the results that might inform public health strategies to increase vaccine uptake.
In this light, I recommend replacing Table 2 in the article with the S1 Table that features comparison between the vaccination groups on the predictor variables. In the SS1 table, the reader can readily discern significant beliefs that might be targeted in campaigns with young people One can also see the importance of family and doctor recommendations regarding the vaccine can be readily seen in this table as well. The overall levels of uncertainty about vaccine effectiveness and safety are also apparent in the descriptive results in this survey.
Based on this, I would like the you to expand on strategic audiences and campaign strategies that might be developed in light of these results. For instance, medical practitioners could be informed on how important their recommendations are for increasing vaccine uptake. Alternately, young people could also be advised to specifically ask their primary care medical personnel to talk to them about the safety and efficacy of the HPV vaccines. Parents of young people also appear to be a very important part of the decision-making about whether to get the HPV vaccine. How might parents be influenced to take a more active role in the medical consultation process by medical professionals etc. The authors have one paragraph that briefly lists some possible implications, but I would like to see this discussed in more depth.
I do question the wisdom of collapsing the five-point response scales into the categories of agree, neutral and disagree. This amounts to throwing away explanatory variance. If I were analyzing the data, I would favor a discriminant analysis which would use the predictor variables to predict actual group membership (fully vaccinated, initiated but not completed, unvaccinated and uncertain), This would give us a cumulative idea of how much we are able to reduce prediction error as a result of this multivariate analysis. I am not recommending a reanalysis of the data along these lines, but I would recommend taking out the lines about this data collapse making the results more interpretable, because I do not think that that is the case.
Also recheck line 52 in the document-I think that sentence repeats racial disparities twice.
This is an important research piece and it was a pleasure to read. Best wishes in your future endeavors with this paper.
Author Response
- Your write-up provides a comprehensive test of predictors of HPV vaccine uptake with 44 predictor variables. It includes beliefs, attitudes, sociological variables such as friend and family support, and demographic factors. The sampling for the survey also has good representation of different racial and ethnic groups. Overall, the study is well designed and methodologically sound. It is NIH funded, so it has to be.
One obvious limitation of your survey data the large number of the respondents who were unsure of their vaccination status. Fortunately, the cross-tabulation tables in the supplementary material allow the observer to compare between the different groups across the independent predictor variables and it seems that the "vaccination unknown" group bears some resemblance to the "known unvaccinated group".
The results of this study provides some useful information for public health practitioners trying to develop strategies to increase vaccine uptake of the HPV vaccine. When I read a manuscript such as this, I am primarily interested in the public health practice implications of the research. In the current piece, I was primarily interested in the results that might inform public health strategies to increase vaccine uptake.
In this light, I recommend replacing Table 2 in the article with the S1 Table that features comparison between the vaccination groups on the predictor variables. In the SS1 table, the reader can readily discern significant beliefs that might be targeted in campaigns with young people. One can also see the importance of family and doctor recommendations regarding the vaccine can be readily seen in this table as well. The overall levels of uncertainty about vaccine effectiveness and safety are also apparent in the descriptive results in this survey.
Response: Thank you for your suggestion. We have added S1 Table to the main text (see Table 3). The text reads: “The descriptive patterns shown in Table 3, our findings highlight opportunities to improve for public health messaging, particularly the need to strengthen physician recommendations, address concerns about vaccine safety and side effects, and emphasize both personal susceptibility to HPV and the preventive benefits of vaccination, which remain underrecognized among a sizable proportion of respondents.” (Lines 405-409).
- Based on this, I would like you to expand on strategic audiences and campaign strategies that might be developed in light of these results. For instance, medical practitioners could be informed on how important their recommendations are for increasing vaccine uptake. Alternately, young people could also be advised to specifically ask their primary care medical personnel to talk to them about the safety and efficacy of the HPV vaccines. Parents of young people also appear to be a very important part of the decision-making about whether to get the HPV vaccine. How might parents be influenced to take a more active role in the medical consultation process by medical professionals etc. The authors have one paragraph that briefly lists some possible implications, but I would like to see this discussed in more depth.
Response: We have discussed additional strategies to increase vaccine uptake based on the study results focusing on the health care providers and families. The added text reads “ A majority of respondents were aware of HPV and its link to cancer, yet a substantial proportion expressed concerns about vaccine safety and perceived low personal risk. These gaps emphasize the critical role of healthcare providers in recommending the HPV vaccine and addressing persistent concerns about safety and perceived risk. Encouraging young adults to initiate conversations with medical professionals about HPV vaccination may further strengthen engagement and informed decision-making. Asparents could play a significant role in the vaccination process, empowering parents through education and provider outreach may further promote their active participation in vaccination decisions and support higher uptake. (Line 410-418).”
- I do question the wisdom of collapsing the five-point response scales into the categories of agree, neutral and disagree. This amounts to throwing away explanatory variance. If I were analyzing the data, I would favor a discriminant analysis which would use the predictor variables to predict actual group membership (fully vaccinated, initiated but not completed, unvaccinated and uncertain), This would give us a cumulative idea of how much we are able to reduce prediction error as a result of this multivariate analysis. I am not recommending a reanalysis of the data along these lines, but I would recommend taking out the lines about this data collapse making the results more interpretable, because I do not think that that is the case.
Response:.” (Line166-168). Thank you. We have removed the text regarding the benefit of collapsing the response for improving interpretability. It reads “…..beliefs about susceptibility and se-verity of HPV infection, normative beliefs, self-efficacy and cues to action for HPV vaccina-tion on a five-point Likert scale as ‘strongly disagree’, ‘disagree’, ‘neutral’, ‘agree’ ‘strongly agree’. Responses of ‘strongly disagree’ collapsed with ‘disagree’ and ‘strongly agree’ with ‘agree’ during data analysis. (see lines 166-168).
4 Also recheck line 52 in the document-I think that sentence repeats racial disparities twice.
Response: Thank you for catching these, we have edited the text removing the duplicate. (Line 51).
Reviewer 4 Report
Comments and Suggestions for Authors
Dear Authors,
The paper is very interesting and at the same time simple.
It will be very useful for preventing HPV infection.
It is technically very good, and methodologically as well.
I would like to make a few comments.
If a sample size study was conducted, it should be indicated, and if not, the statistical power obtained with this sample should be described.
According to Table 1, the groups are sociodemographically distinct. This must be taken into account.
The statistical analyses are correct, but I haven't seen whether any age differences have been calculated. In this case, they should indicate which test was used.
In the odds ratio graphs, they should indicate that the results are shown to the right of the 1 and to the left of the 1. This will make interpretation easier for readers less familiar with this terminology, and the graph will be self-explanatory.
Otherwise, you have done a great and good job.
Author Response
Comments and Suggestions for Authors
Dear Authors,
The paper is very interesting and at the same time simple.
It will be very useful for preventing HPV infection.
It is technically very good, and methodologically as well.
I would like to make a few comments.
- If a sample size study was conducted, it should be indicated, and if not, the statistical power obtained with this sample should be described.
Response: Thank you. We have added a text explaining the sample size estimating strategy. It reads
“2.6. Sample size: Since this study was part of a larger project evaluating the utility of health behavior theories in explaining factors influencing HPV vaccination using structural equation modeling (SEM), the sample size was determined based on the requirements of the overall study.51 Briefly, the SEM based on the Integrated Health Theory (IHT) included 184 estimated parameters, comprising 39 factor loadings, 65 variances, 25 covariances, and 55 structural paths. A participant-to-parameter ratio of 7 is generally recommended to ensure adequate statistical power for SEM estimation. Accordingly, a minimum of 1,288 participants was required to test the validity of the conceptual frameworks derived from the IHT, Theory of Planned Behavior, and Health Belief Theory. Qualtrics ultimately provided data from 1,306 participants, meeting this requirement.
- According to Table 1, the groups are sociodemographically distinct. This must be taken into account. The statistical analyses are correct, but I haven't seen whether any age differences have been calculated. In this case, they should indicate which test was used.
Response: Thank you. We have tested whether there was significant difference in the mean age of the study participants based on their vaccination status using the Kruskal–Wallis test and presented the results in Table 1. The added text reads “Given the non-normal distribution of age, we assessed age differences across vaccination groups using the Kruskal–Wallis test” (lines 204 & 205). The added text in the result section (Table 1 footnote) reads “Age differed across vaccination groups (Kruskal–Wallis, p < 0.0001)”
- In the odds ratio graphs, they should indicate that the results are shown to the right of the 1 and to the left of the 1. This will make interpretation easier for readers less familiar with this terminology, and the graph will be self-explanatory.
Otherwise, you have done a great and good job.
Response: Thank you for the suggestion. We have added a caption to the supplemental figures clarifying that odds ratios to the right of 1 indicate higher odds of vaccination, while those to the left of 1 indicate lower odds.
Reviewer 5 Report
Comments and Suggestions for Authors
Dubon et al. have conducted a cross-sectional Qualtrics-panel survey of 1,309 Midwestern adolescents and young adults (13–26 years) to assess HPV vaccination status and multilevel predictors of initiation and completion. They report 30.7% fully vaccinated (defined as 2–3 doses), 9.5% one dose, 25% unvaccinated, and 35% unsure of status. Facilitators of completion included HPV vaccine awareness (OR≈2.4), belief in vaccine effectiveness (OR≈1.8), family support (OR≈2.3), and knowing someone with cervical cancer (OR≈1.8). For initiation, belief in vaccine safety (OR≈2.0) and having insurance (OR≈1.9) increased odds, whereas concern about side effects (OR≈0.5) and lack of clinician recommendation (OR≈0.5) reduced both initiation and completion; clinician recommendation and awareness also lowered the likelihood of “unknown” status. Race-stratified models indicated heterogeneous patterns (e.g., safety beliefs most strongly predicted completion among non-Hispanic Black participants; family support and effectiveness beliefs were key among non-Hispanic White participants; knowing someone with cervical cancer consistently facilitated uptake in Hispanic/Latino groups). The authors emphasize implications for multilevel interventions targeting families, clinicians, and access, and acknowledge limitations including self-reported vaccination, cross-sectional design, limited power in some racial/ethnic strata, and absence of information on delivery models (e.g., school-based vs clinic/pharmacy).
General comments
The manuscript addresses an important gap and is, overall, well grounded in prior literature, with a clear presentation of methods and results. That said, several revisions would strengthen accuracy, relevance, and policy applicability. First, describe the delivery context for HPV vaccination in the Midwest (school-based vs opportunistic clinic/pharmacy/primary care), as mode of delivery is a key determinant of uptake and equity. Second, align the framing with current programmatic guidance: note WHO’s elimination framework (90–70–90) and that an 80% uptake target is sub-optimal relative to elimination benchmarks; acknowledge WHO-endorsed single-dose schedules and, where possible, provide sensitivity analyses using ≥1 dose (“up-to-date”) alongside the manuscript’s 2–3 dose “fully vaccinated” definition. Third, discuss practical levers to raise coverage—co-administration/bundling at adolescent visits, earlier initiation at ages 9–10 (endorsed by AAP/ACS) to improve on-time completion, and the rationale (from other pediatric vaccines) that combination products increase coverage and timeliness. Finally, clarify limitations (self-report, cross-sectional design, power in racial/ethnic strata) and emphasize how multilevel, equity-focused interventions (provider recommendation quality, family engagement, improved access/coverage) follow from your findings.
Specific comments
1. The manuscript does not describe how HPV vaccination is delivered in the Midwest (e.g., school-based delivery versus clinic/pharmacy/primary-care opportunistic delivery). This omission matters for interpretation: mode of delivery is a well-established determinant of uptake and completion, and helps explain regional shortfalls relative to targets. I recommend adding brief context on U.S. delivery models and discussing how school-based versus opportunistic delivery could shape initiation, completion, and equity in the Midwest.
Evidence from countries with school-based programs shows consistently high coverage: Australia’s national school program reports ≥1-dose coverage at age 15 of 84.2% (girls) and 81.8% (boys) in 2023; England’s school-aged program achieves high uptake across Year 8–10 cohorts; Norway’s school program has reached ~89% among 16-year-old girls; and Sweden’s school program shows high coverage with strong adherence to recommended intervals. These data—and reviews comparing delivery models—support the contention that school-based vaccination facilitates higher and more equitable uptake than purely opportunistic systems.
Wang J, Herweijer E, Nordqvist Kleppe S, Hartwig S, Velicer C, Koro C, Sundström K. High coverage and adherence to dose intervals of the national school-based HPV vaccination program in Sweden during 2012-2019. Prev Med Rep. 2023 Jul 22;35:102342. doi: 10.1016/j.pmedr.2023.102342. PMID: 37584061; PMCID: PMC10424256.
National Centre for Immunisation Research and Surveillance (NCIRS). Vaccination coverage in adolescents: Vaccination coverage for adolescents in Australia for the calendar year 2023. Sydney: NCIRS; 26 Sep 2025. Available online: https://ncirs.org.au/annual-immunisation-coverage-report-2023-summary/vaccination-coverage-adolescents
UK Health Security Agency (UKHSA). Human papillomavirus (HPV) vaccination coverage in adolescents in England: 2023 to 2024 (Official Statistics; updated 24 Jun 2025). London: UKHSA; 2025. Available online: https://www.gov.uk/government/statistics/human-papillomavirus-hpv-vaccine-coverage-estimates-in-england-2023-to-2024/human-papillomavirus-hpv-vaccination-coverage-in-adolescents-in-england-2023-to-2024
Bjerke RD, Laake I, Feiring B, Aamodt G, Trogstad L. Time trends in HPV vaccination according to country background: a nationwide register-based study among girls in Norway. BMC Public Health. 2021 May 3;21(1):854. doi: 10.1186/s12889-021-10877-8. PMID: 33941126; PMCID: PMC8091748.
Wang J, Ploner A, Sparén P, Lepp T, Roth A, Arnheim-Dahlström L, Sundström K. Mode of HPV vaccination delivery and equity in vaccine uptake: A nationwide cohort study. Prev Med. 2019 Mar;120:26-33. doi: 10.1016/j.ypmed.2018.12.014. Epub 2018 Dec 27. PMID: 30593796.
2. Co-administering HPV vaccine on the same day as other adolescent vaccines should be explicitly recommended to raise uptake and completion. Multi-component “bundling” interventions that train clinicians, surface missed opportunities, and prompt same-day vaccination increased HPV vaccination during well-child visits by ~5 percentage points across 24 pediatric practices, demonstrating fewer missed opportunities when HPV is offered with other vaccines. Real-world quality-improvement work in a safety-net dermatology/STI clinic showed that implementing in-office, same-day HPV vaccination substantially increased initiation and series completion—underscoring that having HPV vaccine available and administered at the index visit is critical to success. Population-level data also show growing U.S. “Tdap-HPV bundling,” with the proportion of Tdap visits that included same-day HPV rising from 22.9% to 39.1% (2014–2018); bundling was associated with higher odds of initiation in younger adolescents, supporting co-administration as a scalable pathway to higher coverage.
Szilagyi PG, Fiks AG, Rand CM, Kate Kelly M, Russell Localio A, Albertin CS, Humiston SG, Grundmeier RW, Steffes J, Davis K, Shone LP, McFarland G, Abney DE, Stephens-Shields AJ. A Bundled, Practice-Based Intervention to Increase HPV Vaccination. Pediatrics. 2025 Feb 1;155(2):e2024068145. doi: 10.1542/peds.2024-068145. PMID: 39756464; PMCID: PMC12285739.
Himeles JR, McKenzie C, Manduca S, Shaw KS, Jones Z, Nagler A, Pomeranz MK, Gutierrez D, Zampella JG. Same-day human papilloma virus vaccination improves vaccine uptake in a dermatology sexually transmitted infection clinic: A quality improvement-based model for improving vaccination rates. J Am Acad Dermatol. 2025 Jun;92(6):1288-1294. doi: 10.1016/j.jaad.2025.01.091. Epub 2025 Feb 4. PMID: 39909346.
Zhu Y, Wu CF, Giuliano AR, Fernandez ME, Ortiz AP, Cazaban CG, Li R, Deshmukh AA, Sonawane K. Tdap-HPV vaccination bundling in the USA: Trends, predictors, and implications for vaccine series completion. Prev Med. 2022 Nov;164:107218. doi: 10.1016/j.ypmed.2022.107218. Epub 2022 Aug 23. PMID: 36007751; PMCID: PMC9691592.
3. In the introduction, the authors state, “The U.S. Centers for Disease Control and Prevention’s (CDC) Advisory Committee on Immunization Practices (ACIP) recommend initiating HPV vaccination at ages 11–12 years.” However, multiple U.S. professional bodies—notably the American Academy of Pediatrics (AAP) and the American Cancer Society (ACS)—now endorse starting HPV vaccination at ages 9–10 because earlier initiation reliably increases on-time completion and overall coverage. Large claims-based and survey analyses show substantially higher series completion by age 13 when vaccination begins at 9–10 versus 11–12. An earlier start creates more opportunities across routine well-child visits before adolescence, reduces missed opportunities at the crowded 11–12-year visit, and allows more flexible scheduling if doses are deferred. Immunogenicity at 9–10 is at least as robust as in older ages, safety is well established, and earlier vaccination decouples cancer prevention from discussions of sexual debut—improving parental acceptance. Importantly, these benefits are observed across payer types and demographic subgroups, supporting ages 9–10 as a practical, equity-enhancing strategy to raise initiation and completion before age 13.
Saxena K, Kathe N, Sardana P, Yao L, Chen YT, Brewer NT. HPV vaccine initiation at 9 or 10 years of age and better series completion by age 13 among privately and publicly insured children in the US. Hum Vaccin Immunother. 2023 Dec 31;19(1):2161253. doi: 10.1080/21645515.2022.2161253. Epub 2023 Jan 11. PMID: 36631995; PMCID: PMC9980633.
Goodman E, Felsher M, Wang D, Yao L, Chen YT. Early Initiation of HPV Vaccination and Series Completion in Early and Mid-Adolescence. Pediatrics. 2023 Mar 1;151(3):e2022058794. doi: 10.1542/peds.2022-058794. PMID: 36843509.
Saxena K, Patterson-Lomba O, Gomez-Lievano A, Zion A, Cunningham-Erves J, Kepka D. Assessing the long-term implications of age 9 initiation of HPV vaccination on series completion by age 13-15 in the US: projections from an age-structured vaccination model. Front Pediatr. 2024 Jun 27;12:1393897. doi: 10.3389/fped.2024.1393897. PMID: 38993325; PMCID: PMC11238570.
4. Although no licensed HPV-containing combination vaccine currently exists, evidence from other pediatric combination vaccines shows that combining antigens increases coverage, improves timeliness, and reduces missed opportunities—supporting the rationale that an HPV-containing combination could meaningfully raise HPV vaccination uptake. U.S. national analyses report that children receiving at least one combination vaccine have substantially higher odds of completing the full series by 24 months and being vaccinated on schedule compared with those receiving only single-antigen products (OR≈2.5 for completion; OR≈2.2 for on-time vaccination). Cohort studies likewise find greater adherence to recommended schedules among recipients of DTaP-based combinations than among recipients of separate components. A recent systematic review and meta-analysis concludes that widely used infant combinations (e.g., DTaP–IPV–Hib, DTaP–HBV–IPV–Hib) are immunogenic, safe, and programmatically advantageous—supporting higher and more timely coverage while reducing visit burden. Real-world evaluations also show that combination schedules decrease the number of vaccination visits, a mechanism plausibly linked to fewer deferrals and better series completion. Taken together, these data suggest that if manufacturers developed and licensed an HPV-containing combination vaccine, co-formulation could increase initiation and completion by simplifying logistics, minimizing injections per visit, and shrinking missed opportunities at adolescent encounters.
Kurosky SK, Davis KL, Krishnarajah G. Effect of combination vaccines on completion and compliance of childhood vaccinations in the United States. Hum Vaccin Immunother. 2017 Nov 2;13(11):2494-2502. doi: 10.1080/21645515.2017.1362515. Epub 2017 Sep 7. PMID: 28881166; PMCID: PMC5703402.
Loiacono MM, Pool V, van Aalst R. DTaP combination vaccine use and adherence: A retrospective cohort study. Vaccine. 2021 Feb 12;39(7):1064-1071. doi: 10.1016/j.vaccine.2021.01.009. Epub 2021 Jan 20. PMID: 33483215.
Liu B, Cao B, Wang C, Han B, Sun T, Miao Y, Lu Q, Cui F. Immunogenicity and Safety of Childhood Combination Vaccines: A Systematic Review and Meta-Analysis. Vaccines (Basel). 2022 Mar 18;10(3):472. doi: 10.3390/vaccines10030472. PMID: 35335107; PMCID: PMC8954135.
Kim HJ, Park S, Jeong NY, Choi NK. Changes in vaccination practices among infants after the introduction of DTaP-IPV/Hib combination vaccines. Vaccine X. 2024 Apr 9;18:100484. doi: 10.1016/j.jvacx.2024.100484. PMID: 38655547; PMCID: PMC11035106.
5. In the introduction, the authors state, ‘Although coverage in some states in the Midwest remains higher than the national average, it still falls short of the Healthy People 2030 goal of 80%.’ However, an 80% target is below the WHO elimination framework. Cervical cancer elimination is defined as an incidence <4 per 100,000 women, supported by the 90–70–90 targets by 2030: 90% of girls fully vaccinated with HPV vaccine by age 15; 70% of women screened with a high-performance test by ages 35 and 45; and 90% of women with precancer treated and 90% of those with invasive cancer appropriately managed. Aligning regional goals with the WHO 90–70–90 benchmarks—rather than an 80% vaccine threshold—would better reflect what is required to reach elimination.
World Health Organization (WHO). Cervical Cancer Elimination Initiative. Geneva: WHO; 2025. Available online: https://www.who.int/initiatives/cervical-cancer-elimination-initiative
6. In the Methods (Outcomes), the authors define “fully vaccinated” as receipt of 2–3 doses. However, WHO has endorsed single-dose schedules since 2022 and, as of 2024–2025, expanded single-dose use to additional prequalified products (e.g., Cecolin®), with many countries implementing a 1-dose schedule; notably, England adopted a routine single-dose HPV program in September 2023 using Gardasil 9. Given this shift, I recommend (i) clarifying that the study’s “fully vaccinated” definition reflects historical U.S. schedules during the study period, (ii) adding a sensitivity analysis that classifies “up-to-date” vaccination as ≥1 dose in line with current WHO and national practice, and (iii) discussing how choice of endpoint (2–3 doses vs ≥1 dose) affects interpretation, comparability with contemporary coverage reports, and policy relevance.
World Health Organization (WHO). WHO adds an HPV vaccine for single-dose use (News release; 4 Oct 2024). Geneva: WHO; 2024. Available online: https://www.who.int/news/item/04-10-2024-who-adds-an-hpv-vaccine-for-single-dose-use
UK Health Security Agency (UKHSA). Information on the HPV vaccination from September 2023 (Guidance; updated 15 Sep 2025). London: UKHSA; 2025. Available online: https://www.gov.uk/government/publications/hpv-vaccine-vaccination-guide-leaflet/information-on-the-hpv-vaccination-from-september-2023
Minor revisions
Lines 2-4, Title, "HPV Vaccination in the U.S. Midwest: Barriers and Facilitators of Initiation and Completion in Adolescents and Young Adults"
Lines 9-26, Abstract, "HPV vaccination uptake among adolescents and young adults in the United States remains suboptimal, and coverage in the Midwest falls short of the Healthy People 2030 goal. We aimed to identify facilitators and barriers to HPV vaccination in the Midwest. In a cross-sectional Qualtrics-panel survey of 1,309 individuals aged 13–26 years, 397 (30.7%) were fully vaccinated (defined as 2–3 doses), 124 (9.5%) had received one dose, 324 (25.0%) were unvaccinated, and 461 (35.2%) were unsure of their status. Awareness of HPV vaccines (OR 2.4; 95% CI 1.6–3.6), belief in vaccine effectiveness (OR 1.8; 95% CI 1.1–2.9), family support (OR 2.3; 95% CI 1.4–3.8), and knowing someone with cervical cancer (OR 1.8; 95% CI 1.2–2.7) were associated with higher odds of full vaccination. For initiation (≥1 dose), belief in vaccine safety (OR 2.0; 95% CI 1.0–3.9) and health insurance (OR 1.9; 95% CI 1.0–3.5) increased odds. Concerns about side effects (OR 0.5; 95% CI 0.3–0.8) and lack of clinician recommendation (OR 0.5; 95% CI 0.3–0.8 for completion; OR 0.5; 95% CI 0.2–0.9 for initiation) decreased both initiation and completion; clinician recommendation and awareness also reduced the likelihood of unknown status. Race-stratified analyses suggested heterogeneity in predictors across racial/ethnic groups. Findings support multilevel strategies—strengthening routine provider recommendations, family engagement, and safety/effectiveness communication, alongside improved access/coverage—to increase initiation and completion in the Midwest. Limitations include self-reported vaccination, cross-sectional design, and limited power for some subgroups."
Lines 30-43, Introduction, "Human papillomavirus (HPV) is the most common sexually transmitted infection in the United States, with an estimated prevalence of ~40% among individuals aged 15–24 years [1]. In 2018, approximately 42 million Americans were infected with HPV types associated with cancer and other diseases [2–6]. Most infections clear spontaneously [7], but non-oncogenic types can cause benign lesions such as genital warts, while oncogenic types are the principal cause of cervical and anal cancers and are implicated in vaginal (~75%), vulvar (~69%), penile (~63%), and oropharyngeal (~70%) cancers [5,7–9]. These HPV-related cancers are largely preventable through vaccination [2,10–12]. The U.S. Advisory Committee on Immunization Practices (ACIP) recommends routine HPV vaccination at ages 11–12 years, with eligibility extending through age 26; individuals initiating before age 15 should receive a 2-dose series (6–12 months apart), and those starting at ≥15 years or who are immunocompromised should receive a 3-dose series [13].
"
Lines 246-261, Discussion, "This study identifies individual, interpersonal, and organizational barriers and facilitators of HPV vaccination among adolescents and young adults in the U.S. Midwest—factors previously reported elsewhere but not comprehensively described in this region. We also observed race-specific patterns in barriers and facilitators. Notably, only one-third of participants were fully vaccinated, well below recent national estimates for adolescents (≈78%) and young adults (≈47%), and far short of the Healthy People 2030 goal of 80%.
Facilitators associated with higher odds of full vaccination included HPV vaccine awareness, belief in vaccine effectiveness, provider recommendation, family support, and knowing someone with cervical cancer. Belief in vaccine safety was associated with vaccine initiation (≥1 dose). Provider recommendation and awareness of HPV-related cancers also reduced the likelihood of unknown vaccination status."
Lines 313-349, Discussion, "Our study enrolled a large, demographically balanced sample that we consider broadly representative of the U.S. Midwest. Unlike many prior analyses that assess adolescents and young adults separately, we examined both groups together to provide a more comprehensive view of coverage and predictors, and we conducted racial/ethnic–stratified analyses. Limitations include potential selection bias inherent to the cross-sectional design; self-reported vaccination not validated against records (information/measurement bias); inability to infer causality between knowledge, attitudes, practices, and uptake; limited power for some subgroup comparisons—suggesting the need to oversample underrepresented groups in future work; and restricted generalizability beyond the Midwest.
Our findings highlight actionable opportunities to raise HPV vaccination rates in adolescents and young adults. Communication strategies that increase awareness of HPV-related risks and the vaccine’s safety and effectiveness may address key barriers. Because facilitators extend beyond individual factors, multilevel interventions targeting adolescents/young adults, families, and healthcare providers are warranted. Parent- and provider-focused initiatives that strengthen recommendation quality and educational messaging could increase uptake, and expanding affordable access (e.g., through insurance coverage) may mitigate cost-related hesitancy. These findings can guide population-based interventions that remove barriers and leverage facilitators to improve coverage.
In summary, we identified individual, interpersonal, and organizational determinants of uptake, with patterns that varied across racial and ethnic groups. Concerns about side effects and lack of clinician recommendation limited both initiation and completion, whereas HPV awareness and knowing someone with cervical cancer facilitated both outcomes. Belief in vaccine effectiveness and family support were key facilitators of completion, while belief in vaccine safety, insurance coverage, and perceived seriousness of HPV-related cancers supported initiation. These findings point to the need for multilevel interventions and should be tested in longitudinal and intervention studies to increase initiation and completion."
Lines 350, Conclusions, "Our cross-sectional survey of Midwestern adolescents and young adults found low HPV vaccination uptake and identified multilevel, racial/ethnic–specific barriers (e.g., concerns about side effects, lack of provider recommendation) and facilitators (e.g., vaccine awareness, perceived effectiveness and safety, family support, knowing someone with cervical cancer). These findings underscore the need for multicomponent strategies that pair strong, routine provider recommendations with family engagement, safety/effectiveness messaging, and improved access/coverage. Given the study’s reliance on self-reported vaccination, cross-sectional design, and limited power for some racial/ethnic subgroups, longitudinal and intervention studies are warranted to test targeted approaches and address inequities. Implementing such evidence-informed, multilevel interventions could increase initiation and completion and help the Midwest move toward national and global coverage goals."
Author Response
Dubon et al. have conducted a cross-sectional Qualtrics-panel survey of 1,309 Midwestern adolescents and young adults (13–26 years) to assess HPV vaccination status and multilevel predictors of initiation and completion. They report 30.7% fully vaccinated (defined as 2–3 doses), 9.5% one dose, 25% unvaccinated, and 35% unsure of status. Facilitators of completion included HPV vaccine awareness (OR≈2.4), belief in vaccine effectiveness (OR≈1.8), family support (OR≈2.3), and knowing someone with cervical cancer (OR≈1.8). For initiation, belief in vaccine safety (OR≈2.0) and having insurance (OR≈1.9) increased odds, whereas concern about side effects (OR≈0.5) and lack of clinician recommendation (OR≈0.5) reduced both initiation and completion; clinician recommendation and awareness also lowered the likelihood of “unknown” status. Race-stratified models indicated heterogeneous patterns (e.g., safety beliefs most strongly predicted completion among non-Hispanic Black participants; family support and effectiveness beliefs were key among non-Hispanic White participants; knowing someone with cervical cancer consistently facilitated uptake in Hispanic/Latino groups). The authors emphasize implications for multilevel interventions targeting families, clinicians, and access, and acknowledge limitations including self-reported vaccination, cross-sectional design, limited power in some racial/ethnic strata, and absence of information on delivery models (e.g., school-based vs clinic/pharmacy).
General comments
The manuscript addresses an important gap and is, overall, well grounded in prior literature, with a clear presentation of methods and results. That said, several revisions would strengthen accuracy, relevance, and policy applicability. First, describe the delivery context for HPV vaccination in the Midwest (school-based vs opportunistic clinic/pharmacy/primary care), as mode of delivery is a key determinant of uptake and equity. Second, align the framing with current programmatic guidance: note WHO’s elimination framework (90–70–90) and that an 80% uptake target is sub-optimal relative to elimination benchmarks; acknowledge WHO-endorsed single-dose schedules and, where possible, provide sensitivity analyses using ≥1 dose (“up-to-date”) alongside the manuscript’s 2–3 dose “fully vaccinated” definition. Third, discuss practical levers to raise coverage—co-administration/bundling at adolescent visits, earlier initiation at ages 9–10 (endorsed by AAP/ACS) to improve on-time completion, and the rationale (from other pediatric vaccines) that combination products increase coverage and timeliness. Finally, clarify limitations (self-report, cross-sectional design, power in racial/ethnic strata) and emphasize how multilevel, equity-focused interventions (provider recommendation quality, family engagement, improved access/coverage) follow from your findings.
Specific comments
- The manuscript does not describe how HPV vaccination is delivered in the Midwest (e.g., school-based delivery versus clinic/pharmacy/primary-care opportunistic delivery). This omission matters for interpretation: mode of delivery is a well-established determinant of uptake and completion, and helps explain regional shortfalls relative to targets. I recommend adding brief context on U.S. delivery models and discussing how school-based versus opportunistic delivery could shape initiation, completion, and equity in the Midwest. Evidence from countries with school-based programs shows consistently high coverage: Australia’s national school program reports ≥1-dose coverage at age 15 of 84.2% (girls) and 81.8% (boys) in 2023; England’s school-aged program achieves high uptake across Year 8–10 cohorts; Norway’s school program has reached ~89% among 16-year-old girls; and Sweden’s school program shows high coverage with strong adherence to recommended intervals. These data—and reviews comparing delivery models—support the contention that school-based vaccination facilitates higher and more equitable uptake than purely opportunistic systems.
Wang J, Herweijer E, Nordqvist Kleppe S, Hartwig S, Velicer C, Koro C, Sundström K. High coverage and adherence to dose intervals of the national school-based HPV vaccination program in Sweden during 2012-2019. Prev Med Rep. 2023 Jul 22;35:102342. doi: 10.1016/j.pmedr.2023.102342. PMID: 37584061; PMCID: PMC10424256.
National Centre for Immunisation Research and Surveillance (NCIRS). Vaccination coverage in adolescents: Vaccination coverage for adolescents in Australia for the calendar year 2023. Sydney: NCIRS; 26 Sep 2025. Available online: https://ncirs.org.au/annual-immunisation-coverage-report-2023-summary/vaccination-coverage-adolescents
UK Health Security Agency (UKHSA). Human papillomavirus (HPV) vaccination coverage in adolescents in England: 2023 to 2024 (Official Statistics; updated 24 Jun 2025). London: UKHSA; 2025. Available online: https://www.gov.uk/government/statistics/human-papillomavirus-hpv-vaccine-coverage-estimates-in-england-2023-to-2024/human-papillomavirus-hpv-vaccination-coverage-in-adolescents-in-england-2023-to-2024
Bjerke RD, Laake I, Feiring B, Aamodt G, Trogstad L. Time trends in HPV vaccination according to country background: a nationwide register-based study among girls in Norway. BMC Public Health. 2021 May 3;21(1):854. doi: 10.1186/s12889-021-10877-8. PMID: 33941126; PMCID: PMC8091748.
Wang J, Ploner A, Sparén P, Lepp T, Roth A, Arnheim-Dahlström L, Sundström K. Mode of HPV vaccination delivery and equity in vaccine uptake: A nationwide cohort study. Prev Med. 2019 Mar;120:26-33. doi: 10.1016/j.ypmed.2018.12.014. Epub 2018 Dec 27. PMID: 30593796.
Response: We appreciate this valuable comment and have added new context on HPV vaccine delivery models in the U.S. Midwest in the introduction. (See lines 72-83). The added text reads as “ In the U.S. Midwest, HPV vaccination is administered primarily through pediatricians, family physicians, local health departments, community health centers, and Vaccines for Children (VFC) providers, following CDC and Advisory Committee on Immunization Practices (ACIP) guidelines. Unlike countries with national school-based programs, HPV vaccination in the United States including all Midwestern states is delivered primarily through opportunistic, clinic-based systems. In practice, adolescents receive HPV vaccination during routine primary care or pediatric visits, at local health departments, federally qualified health centers, or pharmacies participating in the VFC program. While some Midwestern states (e.g., Michigan, Minnesota, and Illinois) have school-linked or outreach vaccination events, none operate a universal school-based program. This decentralized approach contributes to variability in initiation and completion across counties and sociodemographic groups, as access depends on healthcare utilization and provider recommendation rather than systematic in-school delivery.”
- Co-administering HPV vaccine on the same day as other adolescent vaccines should be explicitly recommended to raise uptake and completion. Multi-component “bundling” interventions that train clinicians, surface missed opportunities, and prompt same-day vaccination increased HPV vaccination during well-child visits by ~5 percentage points across 24 pediatric practices, demonstrating fewer missed opportunities when HPV is offered with other vaccines. Real-world quality-improvement work in a safety-net dermatology/STI clinic showed that implementing in-office, same-day HPV vaccination substantially increased initiation and series completion—underscoring that having HPV vaccine available and administered at the index visit is critical to success. Population-level data also show growing U.S. “Tdap-HPV bundling,” with the proportion of Tdap visits that included same-day HPV rising from 22.9% to 39.1% (2014–2018); bundling was associated with higher odds of initiation in younger adolescents, supporting co-administration as a scalable pathway to higher coverage.
Szilagyi PG, Fiks AG, Rand CM, Kate Kelly M, Russell Localio A, Albertin CS, Humiston SG, Grundmeier RW, Steffes J, Davis K, Shone LP, McFarland G, Abney DE, Stephens-Shields AJ. A Bundled, Practice-Based Intervention to Increase HPV Vaccination. Pediatrics. 2025 Feb 1;155(2):e2024068145. doi: 10.1542/peds.2024-068145. PMID: 39756464; PMCID: PMC12285739.
Himeles JR, McKenzie C, Manduca S, Shaw KS, Jones Z, Nagler A, Pomeranz MK, Gutierrez D, Zampella JG. Same-day human papilloma virus vaccination improves vaccine uptake in a dermatology sexually transmitted infection clinic: A quality improvement-based model for improving vaccination rates. J Am Acad Dermatol. 2025 Jun;92(6):1288-1294. doi: 10.1016/j.jaad.2025.01.091. Epub 2025 Feb 4. PMID: 39909346.
Zhu Y, Wu CF, Giuliano AR, Fernandez ME, Ortiz AP, Cazaban CG, Li R, Deshmukh AA, Sonawane K. Tdap-HPV vaccination bundling in the USA: Trends, predictors, and implications for vaccine series completion. Prev Med. 2022 Nov;164:107218. doi: 10.1016/j.ypmed.2022.107218. Epub 2022 Aug 23. PMID: 36007751; PMCID: PMC9691592.
Response: Thank you for the suggestion, we expanded the Discussion to highlight recent evidence showing that co-administering HPV with other adolescent vaccine “Lastly, co-administering HPV with other adolescent vaccines through same-day or “bundled” vaccination approaches has been shown to significantly improve initiation and completion rates, reducing missed opportunities and offering a scalable strategy to strengthen routine adolescent immunization.” (Lines 413-420)
- In the introduction, the authors state, “The U.S. Centers for Disease Control and Prevention’s (CDC) Advisory Committee on Immunization Practices (ACIP) recommend initiating HPV vaccination at ages 11–12 years.” However, multiple U.S. professional bodies—notably the American Academy of Pediatrics (AAP) and the American Cancer Society (ACS)—now endorse starting HPV vaccination at ages 9–10 because earlier initiation reliably increases on-time completion and overall coverage. Large claims-based and survey analyses show substantially higher series completion by age 13 when vaccination begins at 9–10 versus 11–12. An earlier start creates more opportunities across routine well-child visits before adolescence, reduces missed opportunities at the crowded 11–12-year visit, and allows more flexible scheduling if doses are deferred. Immunogenicity at 9–10 is at least as robust as in older ages, safety is well established, and earlier vaccination decouples cancer prevention from discussions of sexual debut—improving parental acceptance. Importantly, these benefits are observed across payer types and demographic subgroups, supporting ages 9–10 as a practical, equity-enhancing strategy to raise initiation and completion before age 13.
Saxena K, Kathe N, Sardana P, Yao L, Chen YT, Brewer NT. HPV vaccine initiation at 9 or 10 years of age and better series completion by age 13 among privately and publicly insured children in the US. Hum Vaccin Immunother. 2023 Dec 31;19(1):2161253. doi: 10.1080/21645515.2022.2161253. Epub 2023 Jan 11. PMID: 36631995; PMCID: PMC9980633.
Goodman E, Felsher M, Wang D, Yao L, Chen YT. Early Initiation of HPV Vaccination and Series Completion in Early and Mid-Adolescence. Pediatrics. 2023 Mar 1;151(3):e2022058794. doi: 10.1542/peds.2022-058794. PMID: 36843509.
Saxena K, Patterson-Lomba O, Gomez-Lievano A, Zion A, Cunningham-Erves J, Kepka D. Assessing the long-term implications of age 9 initiation of HPV vaccination on series completion by age 13-15 in the US: projections from an age-structured vaccination model. Front Pediatr. 2024 Jun 27;12:1393897. doi: 10.3389/fped.2024.1393897. PMID: 38993325; PMCID: PMC11238570.
Response: This is an important clarification. We have updated the introduction to reflect that, in addition to the CDC/ACIP recommendation to initiate HPV vaccination at ages 11–12 years, several U.S. professional bodies—including the American Academy of Pediatrics (AAP) and the American Cancer Society (ACS)—now recommend starting vaccination as early as ages 9.(Line 37-41). The added text reads “The U.S. Centers for Disease Control and Prevention’s (CDC) Advisory Committee on Immunization Practices (ACIP), along with the American Academy of Pediatrics (AAP) and the American Cancer Society (ACS) recommend initiating HPV vaccination at ages 11-12 years with the option to begin as early as age 9 and providing catch-up vaccination through age 26.” (Lines 38-41).
- Although no licensed HPV-containing combination vaccine currently exists, evidence from other pediatric combination vaccines shows that combining antigens increases coverage, improves timeliness, and reduces missed opportunities—supporting the rationale that an HPV-containing combination could meaningfully raise HPV vaccination uptake. U.S. national analyses report that children receiving at least one combination vaccine have substantially higher odds of completing the full series by 24 months and being vaccinated on schedule compared with those receiving only single-antigen products (OR≈2.5 for completion; OR≈2.2 for on-time vaccination). Cohort studies likewise find greater adherence to recommended schedules among recipients of DTaP-based combinations than among recipients of separate components. A recent systematic review and meta-analysis concludes that widely used infant combinations (e.g., DTaP–IPV–Hib, DTaP–HBV–IPV–Hib) are immunogenic, safe, and programmatically advantageous—supporting higher and more timely coverage while reducing visit burden. Real-world evaluations also show that combination schedules decrease the number of vaccination visits, a mechanism plausibly linked to fewer deferrals and better series completion. Taken together, these data suggest that if manufacturers developed and licensed an HPV-containing combination vaccine, co-formulation could increase initiation and completion by simplifying logistics, minimizing injections per visit, and shrinking missed opportunities at adolescent encounters.
Kurosky SK, Davis KL, Krishnarajah G. Effect of combination vaccines on completion and compliance of childhood vaccinations in the United States. Hum Vaccin Immunother. 2017 Nov 2;13(11):2494-2502. doi: 10.1080/21645515.2017.1362515. Epub 2017 Sep 7. PMID: 28881166; PMCID: PMC5703402.
Loiacono MM, Pool V, van Aalst R. DTaP combination vaccine use and adherence: A retrospective cohort study. Vaccine. 2021 Feb 12;39(7):1064-1071. doi: 10.1016/j.vaccine.2021.01.009. Epub 2021 Jan 20. PMID: 33483215.
Liu B, Cao B, Wang C, Han B, Sun T, Miao Y, Lu Q, Cui F. Immunogenicity and Safety of Childhood Combination Vaccines: A Systematic Review and Meta-Analysis. Vaccines (Basel). 2022 Mar 18;10(3):472. doi: 10.3390/vaccines10030472. PMID: 35335107; PMCID: PMC8954135.
Kim HJ, Park S, Jeong NY, Choi NK. Changes in vaccination practices among infants after the introduction of DTaP-IPV/Hib combination vaccines. Vaccine X. 2024 Apr 9;18:100484. doi: 10.1016/j.jvacx.2024.100484. PMID: 38655547; PMCID: PMC11035106.
Response: We appreciate the reviewer’s observation regarding the potential role of HPV-containing combination vaccines in improving coverage and timeliness. We have incorporated this point into the Discussion, it reads “Recent advances in HPV-containing combination vaccines highlight a promising strategy to enhance coverage and timeliness; however, their applicability remains limited to future implementation beyond the scope of current delivery systems in the U.S. Midwest.” See lines: 417-420.
- In the introduction, the authors state, ‘Although coverage in some states in the Midwest remains higher than the national average, it still falls short of the Healthy People 2030 goal of 80%.’ However, an 80% target is below the WHO elimination framework. Cervical cancer elimination is defined as an incidence <4 per 100,000 women, supported by the 90–70–90 targets by 2030: 90% of girls fully vaccinated with HPV vaccine by age 15; 70% of women screened with a high-performance test by ages 35 and 45; and 90% of women with precancer treated and 90% of those with invasive cancer appropriately managed. Aligning regional goals with the WHO 90–70–90 benchmarks—rather than an 80% vaccine threshold—would better reflect what is required to reach elimination.
World Health Organization (WHO). Cervical Cancer Elimination Initiative. Geneva: WHO; 2025. Available online: https://www.who.int/initiatives/cervical-cancer-elimination-initiative
Response: We appreciate the reviewer’s point regarding the WHO’s cervical cancer elimination framework and its 90–70–90 targets. We agree that global elimination requires higher thresholds, including ≥90% HPV vaccination coverage by age 15. However, our statement refers specifically to the U.S. Healthy People 2030 objective, which sets a national target of 80% of adolescents up to date with the HPV vaccine series. Given that this benchmark guides federal and state-level monitoring and performance evaluation within the United States, we retained reference to the 80% target to contextualize Midwestern coverage relative to national policy goals. We have, however, included in the discussion revised text that while 80% represents the current U.S. national benchmark, it remains below the WHO elimination threshold, underscoring the need for continued progress to align with global cervical cancer elimination efforts. It reads “Moreover, our study refers to the Healthy People 2030 target of 80% HPV vaccination coverage as the current U.S. benchmark, however, it falls short of the WHO cervical cancer elimination framework’s 90% goal” (Lines 400-403).
- In the Methods (Outcomes), the authors define “fully vaccinated” as receipt of 2–3 doses. However, WHO has endorsed single-dose schedules since 2022 and, as of 2024–2025, expanded single-dose use to additional prequalified products (e.g., Cecolin®), with many countries implementing a 1-dose schedule; notably, England adopted a routine single-dose HPV program in September 2023 using Gardasil 9. Given this shift, I recommend (i) clarifying that the study’s “fully vaccinated” definition reflects historical U.S. schedules during the study period, (ii) adding a sensitivity analysis that classifies “up-to-date” vaccination as ≥1 dose in line with current WHO and national practice, and (iii) discussing how choice of endpoint (2–3 doses vs ≥1 dose) affects interpretation, comparability with contemporary coverage reports, and policy relevance.
World Health Organization (WHO). WHO adds an HPV vaccine for single-dose use (News release; 4 Oct 2024). Geneva: WHO; 2024. Available online: https://www.who.int/news/item/04-10-2024-who-adds-an-hpv-vaccine-for-single-dose-use
UK Health Security Agency (UKHSA). Information on the HPV vaccination from September 2023 (Guidance; updated 15 Sep 2025). London: UKHSA; 2025. Available online: https://www.gov.uk/government/publications/hpv-vaccine-vaccination-guide-leaflet/information-on-the-hpv-vaccination-from-september-2023
Response: We appreciate the reviewer’s thoughtful comment highlighting the recent WHO guidance endorsing single-dose HPV vaccination schedules and national program changes, such as the adoption of single-dose regimens in England.
We state in the introduction that our definition of “fully vaccinated” (receipt of 2–3 doses) reflects the U.S. Advisory Committee on Immunization Practices (ACIP) recommendations that are still in effect and in the U.S. context ≥1-dose coverage is not currently recognized as “up to date”. Hence, we performed the sensitivity analysis redefining vaccination status as ≥1 dose , where we observed that the model retained the same 12 predictors and similarly to our main findings, awareness of the HPV vaccine, family support and knowing someone with cervical cancer increased the odds if HPV vaccination with ≥1-dose compared to unvaccinated participants. Lack of clinician recommendations and concerns about HPV vaccine side effects decreased the odds of ≥1-dose vaccination. Moreover, Belief in vaccine safety, HPV related cancer awareness, and disbelief in being to young to receive the vaccine decreased the odds of unknown vaccination status.
We have clarified that our operational definition reflects the US current guidance of HPV vaccination. It reads: “Our operational definition of “fully vaccinated” reflects the U.S. HPV vaccination schedule in place during the study period.” See lines 261-262.
We added a paragraph in the discussion that addresses this issue and highlights the practical implications of this shift in recommendations. It now reads: “Although the World Health Organization and other countries have adopted single-dose HPV vaccination schedules, the United States has not yet approved a single-dose regimen; therefore, our findings based 2–3-dose endpoints remain directly relevant to the current U.S. vaccination policy and reporting framework. Nonetheless, future adoption of a single-dose schedule could help address persistent barriers to vac-cine initiation and series completion in the U.S.” Lines 420-425.
Minor revisions
- Lines 2-4, Title, "HPV Vaccination in the U.S. Midwest: Barriers and Facilitators of Initiation and Completion in Adolescents and Young Adults"
Response: We shortened the title. It reads “HPV Vaccination in the U.S. Midwest: Barriers and Facilitators of Initiation and Completion in Adolescents and Young Adults”
Lines 9-26, Abstract, "HPV vaccination uptake among adolescents and young adults in the United States remains suboptimal, and coverage in the Midwest falls short of the Healthy People 2030 goal. We aimed to identify facilitators and barriers to HPV vaccination in the Midwest. In a cross-sectional Qualtrics-panel survey of 1,309 individuals aged 13–26 years, 397 (30.7%) were fully vaccinated (defined as 2–3 doses), 124 (9.5%) had received one dose, 324 (25.0%) were unvaccinated, and 461 (35.2%) were unsure of their status. Awareness of HPV vaccines (OR 2.4; 95% CI 1.6–3.6), belief in vaccine effectiveness (OR 1.8; 95% CI 1.1–2.9), family support (OR 2.3; 95% CI 1.4–3.8), and knowing someone with cervical cancer (OR 1.8; 95% CI 1.2–2.7) were associated with higher odds of full vaccination. For initiation (≥1 dose), belief in vaccine safety (OR 2.0; 95% CI 1.0–3.9) and health insurance (OR 1.9; 95% CI 1.0–3.5) increased odds. Concerns about side effects (OR 0.5; 95% CI 0.3–0.8) and lack of clinician recommendation (OR 0.5; 95% CI 0.3–0.8 for completion; OR 0.5; 95% CI 0.2–0.9 for initiation) decreased both initiation and completion; clinician recommendation and awareness also reduced the likelihood of unknown status. Race-stratified analyses suggested heterogeneity in predictors across racial/ethnic groups. Findings support multilevel strategies—strengthening routine provider recommendations, family engagement, and safety/effectiveness communication, alongside improved access/coverage—to increase initiation and completion in the Midwest. Limitations include self-reported vaccination, cross-sectional design, and limited power for some subgroups."
Response: Thank you for the suggested edits. We have incorporated the suggested edits. Please see lines 8-25.
- Lines 30-43, Introduction, "Human papillomavirus (HPV) is the most common sexually transmitted infection in the United States, with an estimated prevalence of ~40% among individuals aged 15–24 years [1]. In 2018, approximately 42 million Americans were infected with HPV types associated with cancer and other diseases [2–6]. Most infections clear spontaneously [7], but non-oncogenic types can cause benign lesions such as genital warts, while oncogenic types are the principal cause of cervical and anal cancers and are implicated in vaginal (~75%), vulvar (~69%), penile (~63%), and oropharyngeal (~70%) cancers [5,7–9]. These HPV-related cancers are largely preventable through vaccination [2,10–12]. The U.S. Advisory Committee on Immunization Practices (ACIP) recommends routine HPV vaccination at ages 11–12 years, with eligibility extending through age 26; individuals initiating before age 15 should receive a 2-dose series (6–12 months apart), and those starting at ≥15 years or who are immunocompromised should receive a 3-dose series [13]."
Response: Thank you for the suggested edits. We have incorporated them. Please see lines 29-39.
- Lines 246-261, Discussion, "This study identifies individual, interpersonal, and organizational barriers and facilitators of HPV vaccination among adolescents and young adults in the U.S. Midwest—factors previously reported elsewhere but not comprehensively described in this region. We also observed race-specific patterns in barriers and facilitators. Notably, only one-third of participants were fully vaccinated, well below recent national estimates for adolescents (≈78%) and young adults (≈47%), and far short of the Healthy People 2030 goal of 80%. Facilitators associated with higher odds of full vaccination included HPV vaccine awareness, belief in vaccine effectiveness, provider recommendation, family support, and knowing someone with cervical cancer. Belief in vaccine safety was associated with vaccine initiation (≥1 dose). Provider recommendation and awareness of HPV-related cancers also reduced the likelihood of unknown vaccination status."
Lines 313-349, Discussion, "Our study enrolled a large, demographically balanced sample that we consider broadly representative of the U.S. Midwest. Unlike many prior analyses that assess adolescents and young adults separately, we examined both groups together to provide a more comprehensive view of coverage and predictors, and we conducted racial/ethnic–stratified analyses. Limitations include potential selection bias inherent to the cross-sectional design; self-reported vaccination not validated against records (information/measurement bias); inability to infer causality between knowledge, attitudes, practices, and uptake; limited power for some subgroup comparisons—suggesting the need to oversample underrepresented groups in future work; and restricted generalizability beyond the Midwest. Our findings highlight actionable opportunities to raise HPV vaccination rates in adolescents and young adults. Communication strategies that increase awareness of HPV-related risks and the vaccine’s safety and effectiveness may address key barriers.
Because facilitators extend beyond individual factors, multilevel interventions targeting adolescents/young adults, families, and healthcare providers are warranted. Parent- and provider-focused initiatives that strengthen recommendation quality and educational messaging could increase uptake, and expanding affordable access (e.g., through insurance coverage) may mitigate cost-related hesitancy. These findings can guide population-based interventions that remove barriers and leverage facilitators to improve coverage.
In summary, we identified individual, interpersonal, and organizational determinants of uptake, with patterns that varied across racial and ethnic groups. Concerns about side effects and lack of clinician recommendation limited both initiation and completion, whereas HPV awareness and knowing someone with cervical cancer facilitated both outcomes. Belief in vaccine effectiveness and family support were key facilitators of completion, while belief in vaccine safety, insurance coverage, and perceived seriousness of HPV-related cancers supported initiation. These findings point to the need for multilevel interventions and should be tested in longitudinal and intervention studies to increase initiation and completion."
Response: Thank you so much for the suggested edits; we have incorporated them into the revised submission. Please see lines 278-281,286-290,350-351,355-358 and 375-379.
- Lines 350, Conclusions, "Our cross-sectional survey of Midwestern adolescents and young adults found low HPV vaccination uptake and identified multilevel, racial/ethnic–specific barriers (e.g., concerns about side effects, lack of provider recommendation) and facilitators (e.g., vaccine awareness, perceived effectiveness and safety, family support, knowing someone with cervical cancer). These findings underscore the need for multicomponent strategies that pair strong, routine provider recommendations with family engagement, safety/effectiveness messaging, and improved access/coverage. Given the study’s reliance on self-reported vaccination, cross-sectional design, and limited power for some racial/ethnic subgroups, longitudinal and intervention studies are warranted to test targeted approaches and address inequities. Implementing such evidence-informed, multilevel interventions could increase initiation and completion and help the Midwest move toward national and global coverage goals."
Response: Thank you for the suggested edits. We have revised the conclusion paragraph in the discussion following your suggestions. Please see lines 437-448.